# Oceanic climate changes threaten the sustainability of Asia's water tower

Qiang Zhang[1,2,10,11 ✉], Zexi Shen[1,2,11 ✉], Yadu Pokhrel[3], Daniel Farinotti[4,5], Vijay P. Singh[6,7,8], Chong-Yu Xu[9], Wenhuan Wu[1,2] & Gang Wang[1,2]

Water resources sustainability in High Mountain Asia (HMA) surrounding the Tibetan Plateau (TP)—known as Asia's water tower—has triggered widespread concerns because HMA protects millions of people against water stress[1,2]. However, the mechanisms behind the heterogeneous trends observed in terrestrial water storage (TWS) over the TP remain poorly understood. Here we use a Lagrangian particle dispersion model and satellite observations to attribute about 1 Gt of monthly TWS decline in the southern TP during 2003–2016 to westerlies-carried deficit in precipitation minus evaporation (PME) from the southeast North Atlantic. We further show that HMA blocks the propagation of PME deficit into the central TP, causing a monthly TWS increase by about 0.5 Gt. Furthermore, warming-induced snow and glacial melt as well as drying-induced TWS depletion in HMA weaken the blocking of HMA's mountains, causing persistent northward expansion of the TP's TWS deficit since 2009. Future projections under two emissions scenarios verified by satellite observations during 2020–2021 indicate that, by the end of the twenty-first century, up to 84% (for scenario SSP245) and 97% (for scenario SSP585) of the TP could be afflicted by TWS deficits. Our findings indicate a trajectory towards unsustainable water systems in HMA that could exacerbate downstream water stress.

As a vital source of meltwater from seasonal snowpack and thousands of glaciers, HMA surrounding the TP, also known as Asia's water tower, protects approximately 800 million people against water stress[1,2]. Rapidly growing Asian economies that house the largest populations of the world, particularly China, India and Pakistan, rely heavily on the meltwater from HMA and are hence particularly vulnerable to increasing water stress owing to changing freshwater availability across the region[1]. Past decades witnessed exacerbated water stress in these countries by virtue of declining regional TWS[3–7] and rising water demands[1,8,9]. These supply–demand imbalances have been increasingly undermining regional food security and sociopolitical stability, which are expected to deteriorate under a warming climate[10–14]. As a drought-resilient source of water, meltwater from glaciers help mitigate water stress for downstream areas[1]. However, this water source is not sustainable and accelerating glacier mass loss may lead to a cascade of unintended and detrimental environmental and ecological consequences[15–20].

Under a warming climate, water availability in HMA is increasingly being affected by accelerating ice mass loss[3,8,19,20] and warming-induced variations in water availability exhibit high spatial variability[8]. During 2000–2019, HMA's glacier loss accounted for about 19% of the global glacier mass loss of 267 ± 16 Gt per year (ref. [19]). This resulted in a decline in TWS in HMA, reducing water availability and exacerbating

water stress in the Indus and Brahmaputra river basins[2,5,8]. Meanwhile, concurrent droughts have been observed in the Aral, Chu–Issyk-Kul and Balkhash basins[1] owing to PME deficit from the North Atlantic[7]. By contrast, increased precipitation in the Sanjiangyuan—the headwaters of the Yellow, Yangtze and Lancang rivers—has caused increased TWS over the central TP[5,8].

The unprecedented changes in HMA's water systems have been reported in numerous studies[3,8,19,20]. However, the atmospheric mechanisms causing the distinct and regionally varying TWS changes are not well understood[5]. Westerlies and the Indian monsoon bring water vapour recharge for precipitation[21–23], and decreasing precipitation in the Himalayas and increasing precipitation in the Pamirs observed during 1979–2010 were attributed to a weakened Indian monsoon[24] and enhanced westerlies[25,26], respectively. Atmospheric circulations also transmit meteorological droughts from the source regions of water vapour[7,27,28], and mid-latitude westerlies have also been identified as potential drivers for HMA's glacier shrinkage[29]. At the same time, the PME deficit transmitted by the westerlies caused TWS deficits across mid-latitude Eurasia but not in the central TP[7]. Given that only limited water vapour reaches the central TP[26], we hypothesize that the TWS increase over the central TP can be attributed to a blocking of the propagation of PME deficits by HMA's high elevations. Because glacier melting dominated TWS changes in HMA[19,30], we argue that

[1]State Key Laboratory of Earth Surface Processes and Resource Ecology, Beijing Normal University, Beijing, China. [2]Faculty of Geographical Science, Beijing Normal University, Beijing, China. [3]Department of Civil and Environmental Engineering, Michigan State University, East Lansing, MI, USA. [4]Laboratory of Hydraulics, Hydrology and Glaciology (VAW), ETH Zürich, Zürich, Switzerland. [5]Swiss Federal Institute for Forest, Snow and Landscape Research (WSL), Birmensdorf, Switzerland. [6]Department of Biological and Agricultural Engineering, Texas A&M University, College Station, TX, USA. [7]Zachry Department of Civil and Environmental Engineering, Texas A&M University, College Station, TX, USA. [8]National Water and Energy Center, UAE University, Al Ain, United Arab Emirates. [9]Department of Geosciences and Hydrology, University of Oslo, Oslo, Norway. [10]Present address: Advanced Interdisciplinary Institute of Environment and Ecology, Beijing Normal University, Zhuhai, China. [11]These authors contributed equally: Qiang Zhang, Zexi Shen. ✉e-mail: zhangq68@bnu.edu.cn; shenzexi@mail.bnu.edu.cn

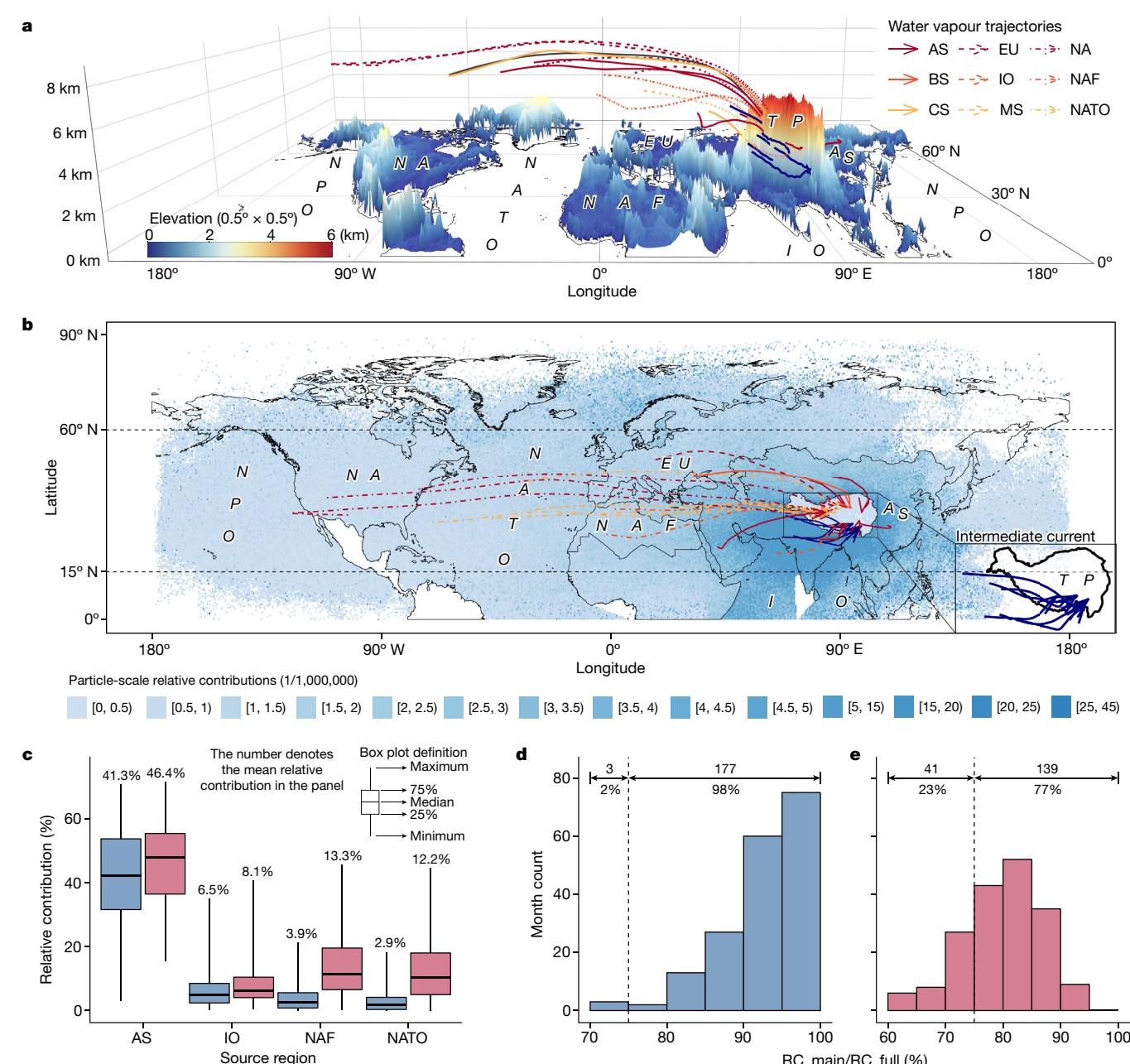

**Fig. 1 | Analysis of water vapour into the TP. a**, The clustered trajectories from source regions to the TP during 2003–2017. **b**, Particle-scale water vapour relative contributions from all source regions to the TP. Intermediate current denotes the current connecting westerlies and the Indian monsoon. **c**, Relative contributions of water vapour from the first to the fourth source regions. **d,e** Combined relative contributions (RC_main) for Asia, India Ocean, North Africa and North Atlantic as compared with the total relative contribution of all considered source regions (RC_full). The month count denotes the number of months when RC_main/RC_full is in each interval. For panels **c**–**e**, blue colour denotes accounting for on-route changes and pink colour denotes disregarding on-route changes, in which 'on-route' means the area between source regions and the TP. The continental map data hereafter are based on country-scale world map data[34]. The TP map data[35] hereafter are acquired from the National Tibetan Plateau Data Center, China. The figure is plotted using R (ref. [36]). The results used to generate the figure are available through Zenodo[37]. AS, Asia (except TP); BS, Black Sea; CS, Caspian Sea; EU, Europe (including Russia); IO, Indian Ocean; MS, Mediterranean Sea; NA, North America; NAF, North Africa; NATO, North Atlantic Ocean; NPO, North Pacific Ocean.

shrinking glaciers and changes in snow cover might have modulated the thermodynamic conditions in HMA and resulted in a damping of HMA's blocking.

## Analysis of water vapour into the TP

By backtracking water vapour using the FLEXPART model (see Methods), we identify the North Atlantic and the Indian Ocean as the main oceanic water vapour sources for the TP (Fig. 1). When accounting for on-route changes (see Methods), we find long-term average relative contributions of about 42% from Asia, about 7% from the Indian Ocean, about 4% from North Africa and about 3% from the North Atlantic (Fig. 1c). Disregarding on-route changes does not markedly alter the outcomes (Fig. 1c). Crucially, during 98% of the study period (77% when disregarding on-route changes), the above four source regions account for more than 75% of the total water vapour input to the TP

(Fig. 1d,e). Distance to the TP has a negative influence on the contributions from source regions to the TP, even though the ocean is the primary continental water source region from a large-scale circulation standpoint. The oceans (for example, the Indian Ocean and the North Atlantic) recharge the water vapour of air particles over the lands (for example, Asia and Africa) that further transit water vapour into the TP (Supplementary Fig. 1) by means of two main trajectories: the first is from the North Atlantic to the western TP and is mediated by the westerlies (Fig. 1a,b), whereas the second is an intermediate current (see Fig. 1a,b) that brings water vapour from the North Atlantic and the Indian Ocean into the southern TP. Relative contributions from Asia are generally the highest owing to the geographical proximity to the TP. When on-route changes are disregarded, relative contributions from the North Atlantic are larger than those from the Indian Ocean (Fig. 1c), raising the question as to whether the TWS in the southern TP mountains is primarily influenced by variations in water vapour input from the Indian Ocean or the North Atlantic.

Our results indicate that the TWS changes along the above trajectories are influenced more prominently by the PME deficits in the North Atlantic than by PME variations in the Indian Ocean and local PME variations. On the basis of monthly relative contributions, we find the water vapour input from the Indian Ocean to be dominant during June–October, whereas that from the North Atlantic dominates mainly during November–May (Supplementary Figs. 2 and 3). Seasonal shifts within the source regions complicate the PME pattern along the routes. Poor correlations are detected between (1) PME across three large-scale subregions (SR1–SR3; see Supplementary Fig. 4a–c) along the trajectories and (2) PME in regions of the North Atlantic and the Indian Ocean (Supplementary Fig. 4d–f). The TWS changes can in turn be attributed to the PME deficit transmitted from the water vapour source regions to the TP[7]. The TWS in SR1–SR3, for example, is strongly related to the PME deficit over the northwest and southeast North Atlantic (correlation coefficients of 0.47–0.60, $P < 0.01$; Supplementary Fig. 4g–i). By contrast, TWS variations in SR1–SR3 are poorly correlated with PME over the Indian Ocean and local PME variations (correlation coefficients between −0.07 and 0.40; Supplementary Figs. 4g–i and 5). Furthermore, relationships between TWS over SR1–SR3 and PME over the Indian Ocean are becoming weaker with decreasing distance to the TP (Supplementary Fig. 4g–i). Instead, PME over the northwest and southeast North Atlantic have persistently notable correlations with TWS over SR1–SR3 (Supplementary Fig. 4g–i). Thus, for further analysis, we focus primarily on the impact on TWS over the TP induced by PME deficit over the North Atlantic.

## Anomalous TWS increase in the central TP

Covariance analysis suggests a distinct water vapour route that transmits PME deficits from the southeast North Atlantic to the TP. This route is characterized by a synchronous decrease in TWS and spatial overlap with eastward water vapour trajectories (Extended Data Fig. 1). The PME deficit from the low-latitude North Atlantic causes a synchronous TWS deficit over mid-latitude (30 °N–60° N) Eurasia[7]. Focusing on regions within 20° N–50° N and considering optimum lags (see Methods), we show high cross-correlations between the PME deficit over the southeast North Atlantic and the TWS changes in 12 out of 14 subregions discerned along the water vapour propagation routes (Fig. 2a–d and Supplementary Fig. 6). This finding is further corroborated by the leading mode of the maximum covariance analysis between (1) the PME deficit over the North Atlantic and the Indian Ocean and (2) TWS across Eurasia, with an explained covariance ratio of 27% (Extended Data Fig. 1a).

Drying over the North Atlantic induced a TWS decrease between mid-latitude Eurasia (20° N–50° N) and the southern TP mountains, but not in the central TP. We detect a decrease in monthly TWS by about 1 Gt (approximately 47%) over the southern TP mountains during 2003–2016, which we attribute to the propagation of a PME deficit from the southeast North Atlantic (Fig. 2e,f), but a synchronous increase in monthly TWS by about 0.5 Gt (approximately 41%) in the central TP (Fig. 2e,f). The latter has been attributed to the TWS gain caused by increased precipitation after a prolonged dry period[5]. Our linear attribution analysis suggests that the increased TWS over the central TP might be related to decreased TWS across the southern TP mountains instead of local increased PME (Extended Data Fig. 2). Furthermore, the cross-correlation analysis suggests that the decreased TWS over the southern TP mountains can be primarily attributed to the propagation of a PME deficit from the southeast North Atlantic (Extended Data Fig. 3m,n). We also detect a notable relation between TWS depletion (for example, glacier mass loss) and the reductions in snow-cover area over the southern TP mountains (Supplementary Fig. 7). In contrast to TWS declines, the reduction in snow-cover area is mainly caused by regional warming (Supplementary Fig. 8). Although glacier and snow meltwater from the southern TP mountains could increase TWS in the central TP[31], such meltwater also replenishes TWS in the basins surrounding the TP. The endorheic character of the central TP differs from the exorheic nature of the basins situated in the south of the TP (for example, the Indus basin in HSR11; Fig. 2b). This difference could potentially allow for the central TP to accumulate meltwater from the southern TP mountains, thus explaining the observed increase in TWS. However, the fact that the TWS over the largest endorheic basin in China (the Tarim basin in HSR12; Fig. 2b) declines despite the same meltwater supply from the southern TP mountains as for the central TP seems to rule out this hypothesis. Moreover, a comparative analysis for the basins surrounding the central TP (HSR8–HSR12) shows that the surrounding regions are directly exposed to the westerlies and intermediate currents that largely terminate at the southern margin of the central TP (Fig. 2b). This means that the TWS changes in the surrounding basins (Extended Data Fig. 3i–l) are consistent with a propagation of the PME deficit by the westerlies from the southeast North Atlantic, whereas this is not the case for the TWS in the central TP (Extended Data Fig. 3o). Because the southern TP mountains belong to HMA, we suggest that the abnormal increase in TWS in the central TP could be linked to changes in blocking effects caused by HMA's high topography. In particular, this topography dampens the propagation of the PME deficit that emerges in the southeast North Atlantic and we hypothesize that a reduction in the propagation of the PME deficit could have resulted in the observed increase in TWS over the central TP. We investigate this hypothesis further in the next section.

## Blocking effects of HMA

FLEXPART simulations suggest that the propagation routes of the PME deficit from the southeast North Atlantic towards the central TP are blocked by HMA's high elevations, causing the routes to split into three trajectories (Figs. 1–3 and Extended Data Fig. 4a). The northern trajectory atmospherically connects the northern TP and southwest TP mountains. The middle trajectory is mainly blocked by the Karakoram and Himalaya mountains, and terminates in the southern parts of the central TP (Fig. 3d and Extended Data Fig. 4a). Finally, the southern trajectory is the intermediate current (Fig. 3d) that runs along the southern side of the Himalaya mountains and moves northward into the TP, along with air currents from the Indian Ocean. It is blocked by the Himalaya and Nyenchen Tanglha mountains and ends in the southeast TP mountains.

The TWS deficits in the southern TP mountains, together with the warming-induced glacier and snow-cover reduction, modulate the thermodynamic conditions of HMA[32,33]. Particularly, we detect marked teleconnections between (1) air temperatures in the southwest TP mountains and the southern North Atlantic and (2) air temperatures in the southeast TP mountains and the northwest Indian Ocean (Extended Data Fig. 5). These two teleconnections are connected by westerlies and the

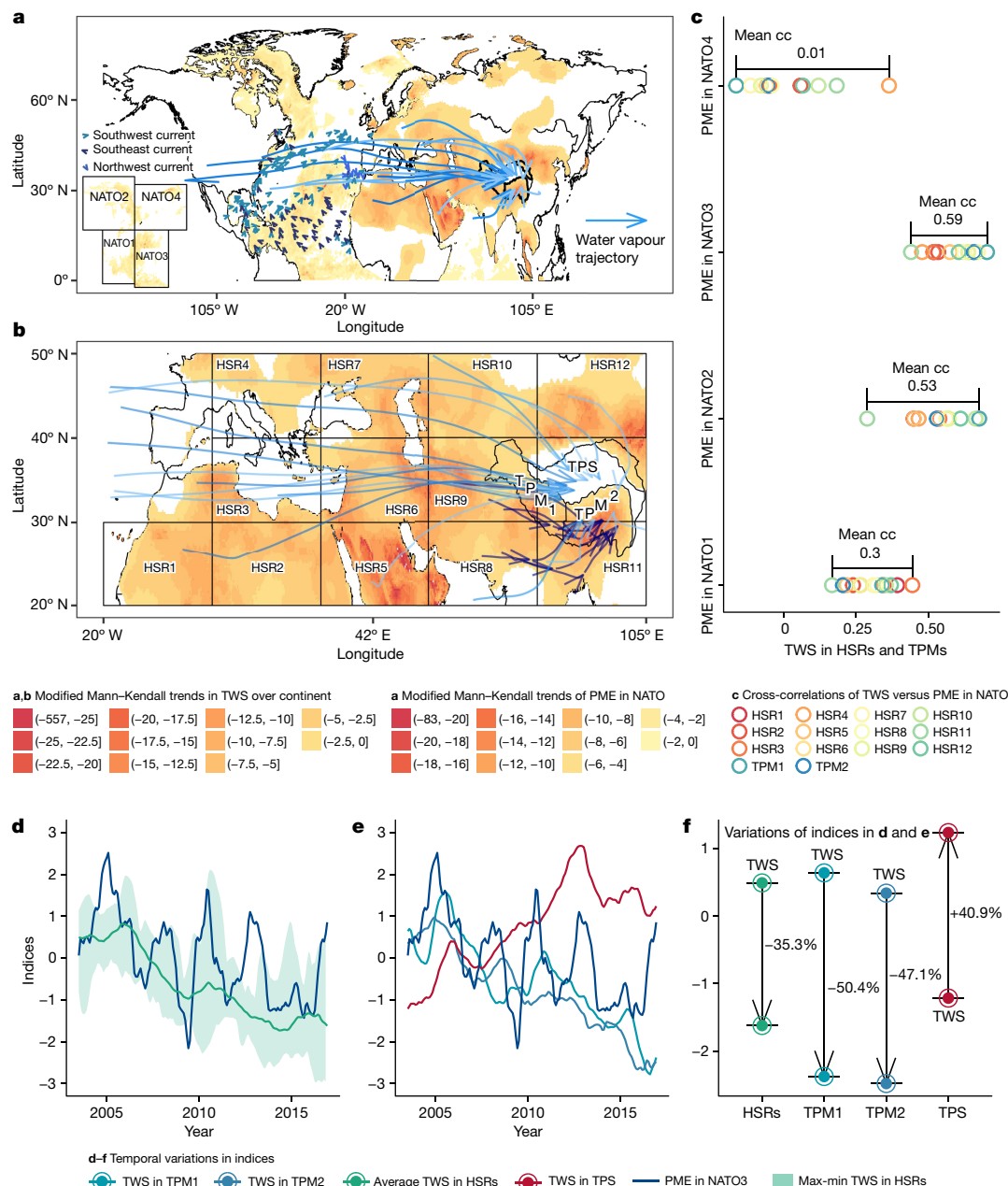

**Fig. 2 | Anomalous TWS increase over the central TP. a,b,** Spatial pattern of the modified Mann–Kendall trends in TWS across Eurasia and PME over oceans. HSR denotes subregions in which TWS has high cross-correlations with PME over NATO3. **c,** Cross-correlations between PME over the North Atlantic and TWS over HSRs across Eurasia and the southern TP mountains (TPM1 and TPM2). Mean cc denotes the average cross-correlation. **d,** Temporal variations in PME over the southeast North Atlantic (NATO3) and average TWS across the HSRs. The coloured band represents the range from maximum to minimum

TWS across the HSRs. **e,** Temporal variations in PME over NATO3 and TWS in the TPMs and the central TP. **f,** Variation in TWS amplitude during 2003–2016. Hereafter, TPM1 and TPM2 refer to southwest and southeast TP mountains, respectively, whereas TPS refers to the central TP surface. The continental map data[34] and the map data of the TP[35] in panels **a** and **b** are from public data sources and plotted using R (ref. [36]). The results used to generate the figure are available through Zenodo[37].

Indian monsoon, respectively (Fig. 1b). Regional warming in the southern TP has led to a reduction in glacier and snow-cover areas. Studies dedicated to atmospheric flows in mountainous terrain[32,33] indicate that the solar radiation affecting snow-free areas leads to convective thermal internal boundary layers, which in turn give rise to positive buoyancy and thus upslope wind. By contrast, negative buoyancy is promoted over snow-covered and ice-covered areas, inducing kataba-tic, downslope winds. Here we suggest that the increased occurrence of upslope winds resulting from a reduction of the snow-cover area could have assisted air flow across the HMA main mountain ridges, thus

weakening their blocking effect. This weakening of the blocking effect could in turn have resulted in an increased propagation of PME deficits into the northern TP, thus contributing to explaining the observed TWS decline in the northern TP (Supplementary Fig. 9). Such a mechanism would also help explain the correlation identified between TWS changes in the northern TP and the variations in temperature and snow-cover area in the southwest TP mountains (Supplementary Fig. 9b,c).

Furthermore, we detect an annual TWS decrease by about 11 Gt (approximately 66% or roughly 13 kg m$^{-3}$) and about 12 Gt (approxi-mately 55% or roughly 12 kg m$^{-3}$) in the southwest and southeast TP

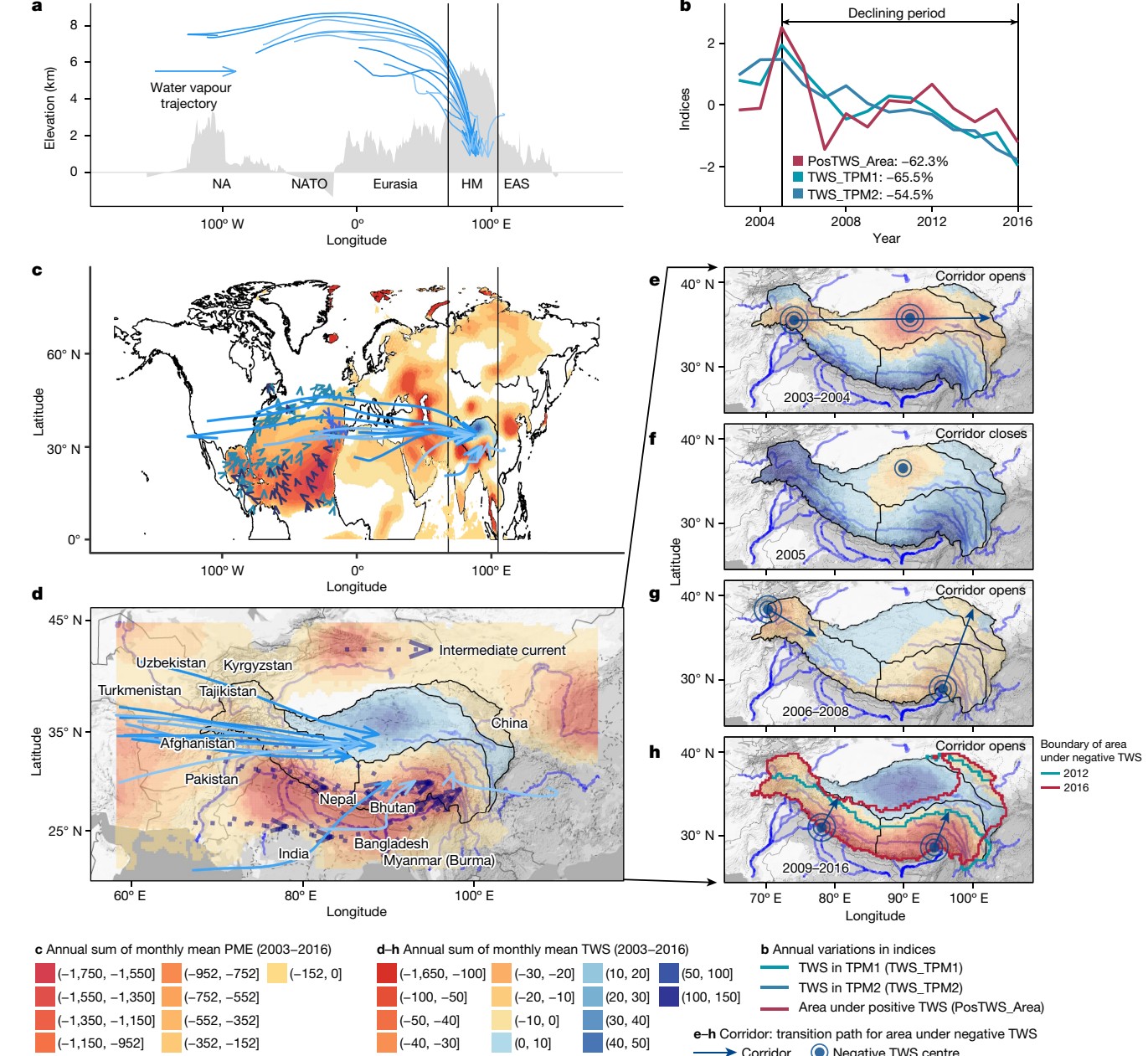

**Fig. 3 | Northward expansion of TWS deficit in the TP. a**, Elevation profile across the mid-latitude Northern Hemisphere and eastward and northward water vapour trajectories. **b**, Annual sum of the standardized trend items of TWS in the southwest and southeast TP mountains and TWS non-deficit area (PosTWS_Area) in the TP during 2003–2016. **c,d** Spatial patterns of annual sums of the monthly mean PME in the North Atlantic and monthly mean TWS across Eurasia during 2003–2016. **e**–**h**, Spatial patterns of the annual sum of TWS

during 2003–2004, 2005, 2006–2008 and 2009–2016, respectively. The continental map data[34] and the map data of the TP[35] in panels **c**–**h** are from public data sources and plotted using R (ref. [36]). The background terrain map is from Google Maps acquired using ggmap[38] in R. The results used to generate the figure are available through Zenodo[37]. EAS, East Asia; HM, Himalayan Mountain.

mountains during 2005–2016, respectively (Fig. 3b). This is partially attributed to snow-cover reduction (Supplementary Fig. 7). Indeed, the annual average snow-cover areas decline by about $11 \times 10^4\,km^2$ (approximately 70%) and about $7 \times 10^4\,km^2$ (approximately 64%) in these two regions, respectively (Extended Data Fig. 6). We suggest that the reductions in snow-cover area and TWS increased upslope winds, thus weakening the blocking effects by assisting air flows (that is, westerlies) to cross HMA. The resulting increase in the propagation of PME deficits into the central TP potentially affects the sustainability of Asia's water tower (Extended Data Fig. 4a). Indeed, the area of the TP

affected by a TWS deficit (that is, the area with TWS ≤ 0) increases by approximately $167 \times 10^4\,km^2$ (about 62%) (Fig. 3b) during 2005–2016.

We also find corridors that act as paths for the area with negative TWS to transit from the southern TP mountains to the central TP (Fig. 3e–h). These corridors roughly align with the westerlies or the Indian monsoon (Fig. 3e–h) and open when there is TWS deficit over the southern TP mountains. During 2003–2016, the propagation of the PME deficit along these corridors caused a TWS deficit expansion across the TP of roughly $224 \times 10^4\,km^2$ (about 73% of the TP) (Fig. 3e–h and Extended Data Fig. 7). In years when the corridor closes (for example, in the year

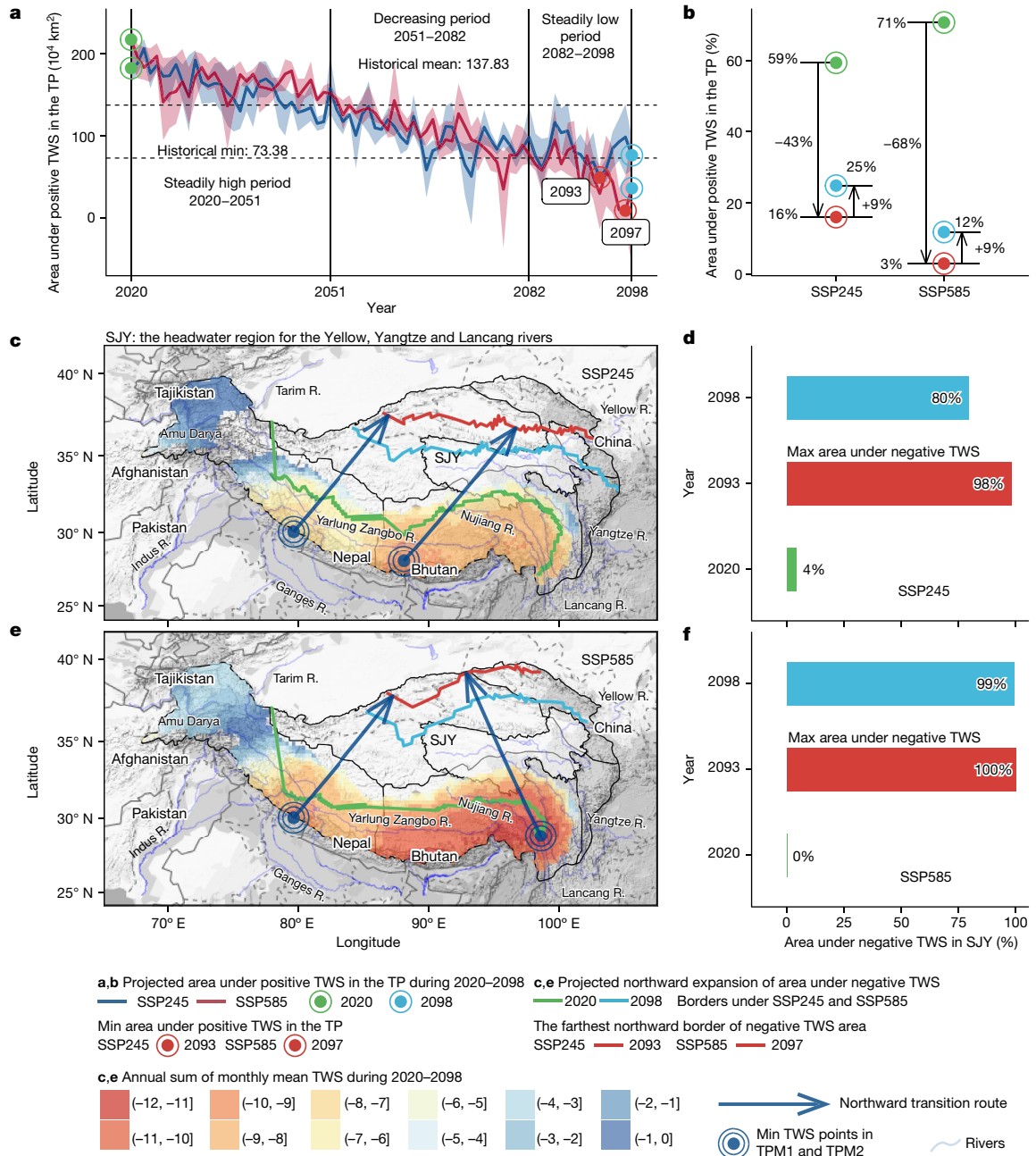

**Fig. 4 | Futural northward expansion of TWS deficit in the TP. a**, Projected annual variation in the area under positive TWS over the TP in future. For each emissions scenario, the coloured band represents the range from maximum to minimum areas determined by the four methods in this study. **b**, Amplitude of area under positive TWS changes over the TP in future. **c,e**, Projected borders of the areas affected by TWS deficit in 2020, 2093 (2097 under SSP585) and 2098. **d,f**, Projected area affected by TWS deficit as a percentage of the total Sanjiangyuan (SJY) area in 2020, 2093 (2097 under SSP585) and 2098. The map data of the TP[35] in panels **c** and **e** are from public data sources and plotted using R (ref. [36]). The background terrain data are from Google Maps acquired using ggmap[38] in R. The results used to generate the figure are available through Zenodo[37].

2005; see Fig. 3f), the related propagation is successfully blocked, preventing a TWS deficit across the central TP. The weakening of the blocking is also visible by the persistent northward expansion of the TWS deficit since 2009−an expansion mediated though middle and southern trajectories (Fig. 3 and Extended Data Figs. 6 and 7). Previous studies have shown that the TWS began to decrease over the central TP in 2013 (refs. [5,30]); however, the cause of such an abrupt TWS decline has not been sufficiently explained. Here we suggest that the persistent northward expansion of the TWS deficit, observed for the TP since 2009, is mainly driven by the westerlies and directly related to the abrupt decrease in 2013 (Figs. 2−3 and Extended Data Figs. 6 and 7).

## Future expansion of the TWS deficit

Influenced by a projected warming of air temperatures over oceans by about 1.8−3.9 °C (Supplementary Fig. 10) and by projected climatic drying over the southeast North Atlantic, continued melting of glaciers and snow might contribute to further enhancing upslope winds and thus to further decreasing the blocking effects of the mountains during the period 2020−2098. Our results indicate that, by the end of the century, this evolution could cause a northward expansion of the area affected by a TWS deficit by up to $258.5 \times 10^4 \, km^2$ (84% of the TP) under SSP245 and by up to $298.6 \times 10^4 \, km^2$ (97% of the TP) under SSP585

(Fig. 4a, b and Extended Data Fig. 8). Note that the average area of the TP that is not affected by TWS deficits during 2020–2051 is greater than the historical average of $137.8 \times 10^4$ km$^2$ (45% of the TP) but then sharply reduces during 2051–2082 (Fig. 4a and Extended Data Fig. 8). The reduction is even more pronounced for the period 2082–2098, in which areas not affected by a TWS deficit are projected to drop to between 27% (SSP245) and 18% (SSP585). This underscores the critical need for reducing carbon emission and for slowing climate warming, otherwise the short-term positive TWS changes in the central TP[8] could reverse. Projected TWS deficits in the southern TP mountains, as well as continuous northward expansion of the TWS deficit over the TP, will probably exacerbate water stress in headwater regions of the main rivers in Asia (Fig. 4c,d). By the end of the century, more than 80% of the Sanjiangyuan will be afflicted by TWS deficit (Fig. 4d,f). These findings imply that the expansion of the area affected by TWS deficit could threaten the sustainability of water supplies from Asia's water tower, exacerbating water stress in the surroundings and affecting millions of people downstream.

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

## Methods

### Environment input data

All data used in this study are accessible from public databases. For the simulation of water vapour transport (see next section) by the FLEXPART v10.4 model[39,40], ERA-Interim atmospheric files covering 61 vertical levels from 0.1 to 1,000 hPa were downloaded and pre-processed by the flex_extract software (https://www.flexpart.eu/downloads). The spatial and temporal resolutions were 0.75° × 0.75° and 6 h, respectively. Monthly precipitation reanalysis data for 2003–2017 were sourced separately from ECMWF Reanalysis 5th Generation (ERA5)[41] and from the Global Precipitation Climatology Centre (GPCC)[42]. These have a spatial resolution of 0.25° × 0.25°. Reanalysis of monthly evaporation and air temperature for 2003–2017 were sourced from ERA5. Monthly multimodel Coupled Model Intercomparison Project Phase 6 (CMIP6)-based precipitation, evaporation and air temperature data for 2003–2099 were acquired from the Center for Environmental Analysis. The considered CMIP6 models included ACCESS-ESM1.5, BCC-CSM2-MR, CanESM5, GFDL-ESM4, IPSL-CM6A-LR, MIROC6, MRI-ESM2.0 and NorESM2-LM under the r1i1p1f1 format. They were coded as CMIP6 models 1 to 8. The TWS anomaly for 2003–2017 was obtained from the GRACE RL05 Mascon dataset[43] (0.5° × 0.5°), whereas the TWS anomaly during 2020–2021 was taken from the GRACE-FO RL06 Mascon dataset[44] (0.25° × 0.25°). The GRACE RL05 data were interpolated to 0.25° × 0.25° using bilinear interpolation[45]. Monthly mean ocean current data for 1993–2019 were acquired from the Ocean Surface Current Analysis Real-time (OSCAR) dataset[46] and have a spatial resolution of 0.33° × 0.33°. The snow-cover data (0.05° × 0.05°) were acquired from the National Snow and Ice Data Center[47]. The continental map data used in the study have been generated on the basis of the continental region groups in the country-scale world map data[34]. There is no conflict on the country boundaries because the map shown in the study is continental. The map data for the TP[35] were acquired from the National Tibetan Plateau Data Center, China.

### Backward simulation of water vapour source regions for the TP

The Lagrangian particle dispersion model FLEXPART v10.4 simulates forward or backward trajectories of air masses (including their water vapour content) by representing a given air mass as a series of small-scale particles[39]. At the particle scale, the balance between evaporation ($e$) and precipitation ($p$) determines the variation in specific humidity (equation (1)). When $e - p > 0$, a particle recharges its water content by evaporation from the surface; when $e - p < 0$, a particle loses its water through precipitation. For a given area, variation in water vapour ($E - P$; uppercase letters denote aggregated quantities) can then be determined as the sum of $e - p$ for all particles in the area (equation (2)),

$$e - p = m\frac{dq}{dt} \tag{1}$$

$$E - P = \sum_{i=1}^{n} (e - p) \tag{2}$$

in which $\frac{dq}{dt}$ denotes the temporal variation in the specific humidity of a particle, $m$ is the mass of the particle and $n$ is the number of particles over the considered area.

We used FLEXPART v10.4 and ERA-Interim climate reanalysis data to perform backward simulations of water vapour trajectories to the TP. This allows us to determine water vapour source regions. The model was then used to perform monthly simulations of trajectories for up to 500,000 particles. The number of the particles depends on the scale of the target region and computational resources[39,48]. Individual simulations were performed at 6-h time steps for the period 2003–2017, enabling the determination of water vapour trajectory from source regions to the TP. Given that the residence time of the water vapour in the atmosphere is approximately 10 days (refs. [49,50]), we performed backward simulations from day 1 to day 10 of each month. A $k$-means method[51] was then used to categorize all particle trajectories into 19 out of 20 main water vapour trajectories based on the spatial locations of particles in individual simulations. For the determination of the number of cluster trajectories, see Supplementary Note 1.

Because FLEXPART v10.4 is driven by ERA-Interim and does not adapt to the more recent ERA5 (this choice is due to the fact that ERA5 data supporting the run of FLEXPART v10.4 is not yet accessible to the public user; for details, see https://www.flexpart.eu/flex_extract/Ecmwf/access.html), a comparison was performed between the simulated total water vapour release and ERA5 and GPCC precipitation over the TP to investigate the sensitivity of simulation results[52]. The total water vapour release for particles residing over the TP was calculated by equation (2) when $E - P < 0$ (ref. [53]) and compared with the standardized regional average precipitation obtained from ERA5 and the GPCC. In both cases, comparisons were performed using correlation analysis. It should be noted that the total water vapour release is not equal to precipitation theoretically but it has the positive effect on the formation of the precipitation[39,48,52]. The total released water vapour over the TP is highly correlated to both the GPCC and ERA5 precipitation, with correlation coefficients between 0.85 and 0.86 ($P$-value < 0.01; see Supplementary Fig. 11). We consider this to be an acceptable accuracy for the FLEXPART results (Supplementary Fig. 11).

On the basis of the initial simulation, 13 primary water vapour source regions were identified: Asia, Indian Ocean, North Africa, North Atlantic, Mediterranean Sea, Europe, Pacific Ocean, Red Sea, Tibet, North America, Caspian Sea, Black Sea and Arctic Ocean. However, because particles can lose or gain water vapour along their track, the water vapour released in the TP can reflect the contribution from both the source regions and the on-route transitions. Thus we calculated the relative contributions in two ways: (1) accounting for or (2) disregarding the on-route changes in water vapour. When accounting for water vapour variations (that is, recharge or loss) along the transitional routes, the considered air particles show positive variations in water vapour inside the source region (equation (4)), because the negative variations inside the source region indicate that the air particles release water vapour instead of acquiring and transporting water vapour from source regions to the TP. Meanwhile, the considered air particles should also be positive variations in water vapour before reaching the TP when accounting for on-route changes (equation (5)).

$$(E - P)_{\text{route}} = \sum_{1}^{r} (e - p) \tag{3}$$

$$(E - P)_{\text{source}} = \sum_{1}^{s} (e - p) > 0 \tag{4}$$

$$(E - P)_{\text{source}} - \text{abs}((E - P)_{\text{route}}) > 0 \tag{5}$$

in which $(E - P)_{\text{route}}$ and $(E - P)_{\text{source}}$ refer to the water vapour variations along the transition routes and over the source regions, respectively. Similarly, $r$ and $s$ refer to the number of particles along the transition routes ($r$) and over the source regions ($s$). Because the air particles might gain or lose water vapour, the absolute variations in water vapour along the transition route should be calculated according to equation (5). For particles originating within the TP, $(E - P)_{\text{source}}$ should be less than zero, thus recharging water vapour in the TP.

After the selection of particles reaching the TP (equations (3)–(5)), relative water vapour contributions from individual source regions were calculated as follows (equations (6) and (7)).

$$(E - P)_i = \sum_{1}^{k_i} (e - p) \ (e - p) < 0, \ i = 1 \text{ to } 13 \tag{6}$$

$$RC_i = \frac{(E-P)_i}{(E-P)_{\text{Tibet}}} \tag{7}$$

in which $(E-P)_i$ refers to the total water vapour released over the TP from particles originating from the $i$th source region, $(E-P)_{\text{Tibet}}$ refers to the total water vapour released in the TP and $RC_i$ is the relative water vapour contribution obtained by the TP from the $i$th source region. When accounting for the on-route variation in water vapour, $k_i$ is the number of selected particles (see equations (3)–(5)) from the $i$th source region. When disregarding water vapour on-route variations, $k_i$ refers to the number of all particles from the $i$th source region.

However, there is another method[52] that determines the relative contribution by dividing the total water vapour release over the target region by the moisture gains over the source regions. To avoid impacts induced by the differences between methods on the results, we also calculated the relative contribution using this method. It suggested that both methods come to the same conclusion that Asia, the Indian Ocean, North Africa and the North Atlantic are the four main water vapour source regions for the TP (Supplementary Fig. 12 and Fig. 1c). However, given that the moisture gains over the source regions are far larger than the total water vapour release in the target region, we applied the first method in this study.

### Standardized trend item of the index
To avoid the impact of periodicity on correlations between variables, a time series of the variable during January 2003 to June 2017 was first smoothed using a moving average as implemented in the R function decompose. The derived trend series of the variable covers the period July 2003 to December 2016. Then, to ensure that different indices can be compared, the trend series of every variable was standardized over the period 2003–2016 by equation (8).

$$X = \begin{cases} \dfrac{x_i - \text{mean}(x)}{\text{sd}(x)}, & x \neq \text{TWS} \\ \dfrac{x_i}{\text{sd}(x)}, & x = \text{TWS} \end{cases} \tag{8}$$

in which $x$ denotes the trend series of the variable, $X$ denotes the standardized version of index $x$, mean($x$) and sd($x$) are the arithmetic average and standard deviation, respectively, of $x$ over the period 2003–2016 and $i$ refers to the $i$th month during 2003–2016. Here the variables include PME, TWS, $T$ or Snowcover, in which Snowcover and $T$ denote the snow-covered area of the southern TP mountains and the air temperature, respectively. The relative variation for a standardized index was calculated by dividing the index variation by the maximum amplitude of all standardized indices (see equation (9)). Furthermore, we also provide mass/area change accordingly for every standardized index.

$$RV_{m,n} = \frac{X_m - X_n}{\max V} \times 100\% \tag{9}$$

in which $X_m$ and $X_n$ denote the values at moments $m$ and $n$ in a time series $X$, $RV_{m,n}$ is the relative variation between $X_m$ and $X_n$, and max$V$ denotes the maximum amplitude of all standardized indices in this study. Because the maximum scale for all standardized indices is in the range $-3$ to 3, we applied max$V$ as 6 here.

Furthermore, to determine the spatial patterns of the trends, we used the modified Mann–Kendall trend analysis as implemented in the R package modifiedmk. This analysis was conducted at the scale of individual grid cells and covers the period 2003–2016.

### Correlation analysis and maximum covariance analysis
Pearson's correlation analysis was used to quantify the relation between standardized indices during 2003–2016. Cross-correlation analysis[54] was applied to evaluate the relation between (1) PME over the North

Atlantic and (2) TWS in the 14 small-scale subregions across mid-latitude Eurasia with consideration of the optimum lags. The optimum lag is defined as the lag when the maximum correlations between two indices are detected. Both the Pearson and the cross-correlation analyses were performed using the R package stats.

Maximum covariance analysis was performed to further verify the covariation between spatial matrixes for any pair of indices. The spatial and temporal coefficients were calculated as follows:

$$\mathbf{L} = \begin{bmatrix} \mathbf{L}_1(1) & \cdots & \mathbf{L}_1(N) \\ \vdots & \ddots & \vdots \\ \mathbf{L}_m(1) & \cdots & \mathbf{L}_m(N) \end{bmatrix} \tag{10}$$

$$\mathbf{R} = \begin{bmatrix} \mathbf{R}_1(1) & \cdots & \mathbf{R}_1(N) \\ \vdots & \ddots & \vdots \\ \mathbf{R}_q(1) & \cdots & \mathbf{R}_q(N) \end{bmatrix} \tag{11}$$

$$\mathbf{C}_{l,r} = \frac{1}{N}\mathbf{L}\mathbf{R}^T = \mathbf{U}\boldsymbol{\Sigma}\mathbf{V}^T \tag{12}$$

$$PC_{l,m} = \mathbf{U}_m^T\mathbf{L} \tag{13}$$

$$PC_{r,q} = \mathbf{V}_q^T\mathbf{R} \tag{14}$$

in which $\mathbf{L}$ and $\mathbf{R}$ refer to the spatial matrixes of (1) PME over both the North Atlantic and the Indian Ocean and (2) TWS across Eurasia, respectively, and indices $m$ and $q$ refer to a given cell in the considered field, whereas the index $k$ refers to the time step. $\mathbf{C}_{l,r}$ is the covariance matrix between $\mathbf{L}$ and $\mathbf{R}$. $\mathbf{U}$ and $\mathbf{V}$ are the spatial modes, respectively, for $\mathbf{L}$ and $\mathbf{R}$. $PC_{l,m}$ and $PC_{r,q}$ are the temporal coefficients for cells $m$ in $\mathbf{L}$ and $q$ in $\mathbf{R}$. The superscript T in equations (10)–(14) denotes the transpose and $\boldsymbol{\Sigma}$ in equation (12) denotes the diagonal matrix. When the maximum correlation analysis is applied to evaluate the relation between $T$ over ocean and land, $\mathbf{L}$ and $\mathbf{R}$ also refer to the spatial matrixes of (1) $T$ over the related oceans and (2) $T$ across Eurasia, respectively.

### Attribution analysis of TWS variation in the central TP
Previous studies reported that both snowmelt in the southern TP mountains and increased precipitation over the central TP contributed to increased TWS in the central TP[5,31]. However, the balance between precipitation and evaporation was not included in those studies. Here we applied two linear regression models to quantify the contributions of both snowmelt in the southern TP mountains and PME changes in the central TP to TWS changes in the central TP. The models are of the form:

$$\text{TWS}_{\text{TPS}} = a_r \times \text{PME}_{\text{TPS}} + b_r \times \text{TWS}_r \tag{15}$$

in which TWS$_r$ refers to the TWS in a given region $r$, which is either the southwest TP mountains (TPM1) or the southeast TP mountains (TPM2), TWS$_{\text{TPS}}$ and PME$_{\text{TPS}}$ denote the TWS and PME in the central TP (TPS), respectively, and $a_r$ and $b_r$ are the regression coefficients determined by linear regression.

### Projections of the TWS non-deficit and deficit areas in the TP
The change in area not affected by TWS deficits (hereafter 'TWS non-deficit area') in the TP was determined by identifying grid cells with annual TWS > 0. On the basis of approximately synchronous variations in the TWS non-deficit area and (1) TWS (Fig. 3b) and (2) snow-covered area (Extended Data Fig. 6) in the southwest and southeast TP mountains, a further linear regression model was developed:

$$A = \sum_r (a_r \times \text{TWS}_r + b_r \times \text{Snowcover}_r) + \varepsilon \tag{16}$$

in which $A$ is the TWS non-deficit area in the TP, $Snowcover_r$ is the snow-covered area over region $r$, $a_r$ and $b_r$ are the regression coefficients and $\varepsilon$ is the model residual.

As there are only very few CMIP6 models that simulate $TWS_r$ and $Snowcover_r$ explicitly, three random forest models (equation (17), $m = 1, 2, 3$) were developed for $TWS_r$ and $Snowcover_r$ projections. The models were first trained and validated on the basis of the GRACE-based TWS or the ERA5-based snow-cover area over region $r$ during 2003–2016 and have the form:

$$D_{r,m} = \mathrm{RF}(Y_k) \qquad (17)$$

in which $Y_k$ denotes the ERA5-based PME (index $k = 1$) over the southeast North Atlantic, ERA5-based $T$ (index $k = 2$) over the southwest TP mountains or ERA5-based $T$ (index $k = 3$) over the southeast TP mountains during 2003–2016. RF denotes the random forest model. The first random forest model was formed by the dependent variable $TWS_r$ ($D_{r,1}$) and independent variable $Y_1$ when $m = 1$. The second random forest model was formed by the dependent variable $TWS_r$ ($D_{r,2}$) and independent variables $Y_1$ and $Y_2$ in southwest TP mountains or $Y_1$ and $Y_3$ in the southeast TP mountains when $m = 2$. The third random forest model was formed by the dependent variable $Snowcover_r$ ($D_{r,3}$) and independent variable $Y_2$ in the southwest TP mountains or $Y_3$ in the southeast TP mountains when $m = 3$. The variable $TWS_r$ was projected with ($m = 1$) and without ($m = 2$) considering $T$, because $TWS_r$ is partially attributed to snow-cover reduction (Supplementary Fig. 7) resulting from local warming (Supplementary Fig. 8).

Equations (16) and (17) were both trained and validated using historical data, reflecting the actual relations between variables (Supplementary Figs. 13–15). These relations were based on the global atmospheric circulation systems (Figs. 1–4) including the westerlies and the Indian monsoon that are generated by the rotation of the Earth and its energy-balance system[55]. Here we project future variations in the TWS non-deficit area in the TP, TWS and snow-covered area in the southern TP mountains using equations (16) and (17) with the CMIP6-modelled data as input.

The distributions of the CMIP6 climatic variables agree well with distributions of the ERA5 climatic variables when $P < 0.05$ (Supplementary Fig. 16). However, given the poor relationships between CMIP6 climate projections from individual model–scenario combinations and ERA5 climatic variables (Supplementary Figs. 17 and 18), (1) average (equation (18)) and (2) weighting sum (equations (19) and (20)) of climatic indices of CMIP6 models were first determined. These can strongly reduce the uncertainties[56] of CMIP6 projections (Supplementary Figs. 19 and 20) and the average and weighting sum of the climatic indices are obtained as follows:

$$\overline{Y}_{k,s} = \frac{\sum_{i=1}^{8}(Y_{k,i,s})}{8} \qquad (18)$$

$$Y_k = f(Y_{k,i,s}, Z_{k,i,s}) \qquad (19)$$

$$W_{k,s} = \sum_i (w_{k,i,s} \times Y_{k,i,s} + c_{k,i,s} \times Z_{k,i,s}) + \varepsilon_{k,s} \qquad (20)$$

in which $s$ denotes the scenario (either SSP245 or SSP585), $\overline{Y}_{k,s}$ denotes the average index for all $Y_{k,i,s}$, $W_{k,s}$ is the weighting sum of $Y_{k,i,s}$ and $Z_{k,i,s}$ denotes either the group of $T$ (index $k = 2$) over the southwest (NATO1) and southeast (NATO3) North Atlantic or the group of $T$ (index $k = 3$) over the four regions in the Indian Ocean (IO1–IO4) derived from the CMIP6 model $i$ under scenario $s$ because air temperature variations over oceans are related to the air temperature over the southern TP mountains (Extended Data Fig. 5). Equation (19) denotes the linear regression model with $Y_k$ (see equation (17)) as the dependent variable and $Y_{k,i,s}$ and $Z_{k,i,s}$ as the independent variables. The weight $w_{k,i,s}$ for $Y_{k,i,s}$,

the weight $c_{k,i,s}$ for $Z_{k,i,s}$ and model residuals $\varepsilon_{k,s}$ in equation (20) are determined by equation (19). For the selection of the index $i$ in equation (19), see Supplementary Note 2. Furthermore, the (1) average and (2) weighting sum of $TWS_r$ or $Snowcover_r$ during 2020–2098 under scenario $s$ are projected by equation (17) with $\overline{Y}_{k,s}$ (equation (18)) and $W_{k,s}$ (equation (20)) as input, and are defined as (1) $\overline{D}_{r,m,s}$ and (2) $WD_{r,m,s}$, respectively.

The annual $A$ during 2020–2098 under scenario $s$ is projected by equation (16) with four combinations of $\overline{D}_{r,m,s}$ and $WD_{r,m,s}$ (Supplementary Fig. 21). The average $A$ (see equation (16)) of these projections ($A_{\mathrm{mean},s}$) is determined by equation (21), which has the form:

$$A_{\mathrm{mean},s} = \frac{\sum_{n=1}^{4} A_{n,s}}{4} \qquad (21)$$

$$\mathrm{Neg}A_{\mathrm{mean},s} = A_{\mathrm{TP}} - A_{\mathrm{mean},s} \qquad (22)$$

in which $A_{n,s}$ denotes $A_{1,s}$ projected by equation (16) with $\overline{D}_{r,1,s}$ and $\overline{D}_{r,3,s}$ as inputs, $A_{2,s}$ projected by equation (16) with $\overline{D}_{r,2,s}$ and $\overline{D}_{r,3,s}$ as inputs, $A_{3,s}$ projected by equation (16) with $WD_{r,1,s}$ and $WD_{r,3,s}$ as inputs or $A_{4,s}$ projected by equation (16) with $WD_{r,2,s}$ and $WD_{r,3,s}$ as inputs. $\mathrm{Neg}A_{\mathrm{mean},s}$ is the average area affected by TWS deficit (hereafter 'TWS deficit area') during 2020–2098 under scenarios $s$, computed as the difference between the $A_{\mathrm{mean},s}$ and the total area of the TP ($A_{\mathrm{TP}}$).

## Evolution of the area under TWS deficit in the TP

Previous studies suggested that the TWS in the central TP is unpredictable owing to the strong interannual variability[5,31]. By contrast, the TWS in the southwest TP and southeast TP are predictable based on the relation proposed in equation (17). Grid-scale TWS in the southern TP mountains is projected by equation (17) when $m = 1$. The centre points of grid cells, at which the annual sum of the monthly average TWS during 2020–2098 is negative (Fig. 4), on the north margin of the southern TP mountains forms the initial northern border of the TWS deficit area in the TP. Because we only study the spatial variation in the TWS deficit area inside the TP, the southern border of the TP is fixed and the area surrounded by the initial northern border and the southern border of the TP is defined as $\mathrm{ini}A$ (Extended Data Fig. 9). The variation in the TWS deficit area ($\Delta A$) was determined by calculating the difference between $\mathrm{Neg}A_{\mathrm{mean},y,s}$ and $\mathrm{ini}A$:

$$\Delta A_{y,s} = \mathrm{ini}A - \mathrm{Neg}A_{\mathrm{mean},y,s} \qquad (23)$$

in which $\mathrm{Neg}A_{\mathrm{mean},y,s}$ is the TWS deficit area in the TP in year $y$ under scenario $s$. When $\mathrm{ini}A > \mathrm{Neg}A_{\mathrm{mean},y,s}$, the area with TWS < 0 shrinks southward by area $|\Delta A_{y,s}|$. By contrast, when $\mathrm{ini}A < \mathrm{Neg}A_{\mathrm{mean},y,s}$, the area with TWS < 0 expands northward by area $|\Delta A_{y,s}|$.

However, $\Delta A$ alone is not enough to determine the areal evolution of the northern border of the TWS < 0 area. Given the stable radial expansion pattern of the TWS < 0 area with two negative centres under the minimum annual TWS sums in the southwest and southeast TP mountains since 2009 (Fig. 3 and Extended Data Fig. 6), future negative centres in these regions are first identified as the cells with the minimum annual sum of the monthly average TWS during 2020–2098 (see Fig. 4c,e) under SSP245 and SSP585 scenarios.

With the aforementioned centres as starting points, we determine the evolving paths as lines between the points forming the initial northern border and the centres in the southern TP mountains (Extended Data Fig. 9). Along the evolving paths, the evolving directions of points forming the initial northern border are thus determined as either northward (for $\mathrm{ini}A < \mathrm{Neg}A_{\mathrm{mean},y,s}$) or southward (for $\mathrm{ini}A > \mathrm{Neg}A_{\mathrm{mean},y,s}$) inside the TP. Furthermore, by setting the expansion steps $\Delta d$ as 0°, 0.25°, 0.5°, 0.75°, 1°, 2°, 3°, 4°, 5°, 6° and 7°, a fitted line between the pairs of $\Delta d$ and $\Delta A$ is identified in the scatter plot (Extended Data Fig. 9). For a given $\Delta A_{y,s}$,

determined by equation (23), a specific $\Delta d$ could be derived by looking up the scatter plot (Extended Data Fig. 9) using the interpolation method. Once $\Delta d$ was determined, by moving all of the points forming the initial northern border along the evolving directions, the northern border of the TWS deficit area inside the TP for a specific year and a specific scenario could be determined (Extended Data Fig. 9). We verify this area expansion model by the observed TWS deficit areas during 2009–2016 and find overlapping ratios of 75–91% (Supplementary Fig. 22). Performing the same test for 2020–2021 results in overlapping ratios of 75–86% (Supplementary Fig. 23). We considered these results to be sufficiently accurate for our analysis.

## Data availability

All data used in this study were obtained from the public database as described in the 'Environment input data' section. The results used to generate the main figures have been deposited in the public repository Zenodo (https://doi.org/10.5281/zenodo.6790243)[37].

## Code availability

All codes used in this study have been deposited in the public repository Zenodo (https://doi.org/10.5281/zenodo.6790243)[37].

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

**Acknowledgements** Q.Z., Z.S., W.W. and G.W. acknowledge the support from the China National Key R&D Program (grant no. 2019YFA0606900), the National Natural Science Foundation of China (grant no. 42041006) and the Engineering Research Center for Water Resources & Ecological Water in Cold and Arid Regions of Xinjiang Uygur Autonomous Region, China (grant no. 2020.A-003). Y.P. acknowledges the support from the National Science Foundation (grant no. 1752729). D.F. acknowledges the support from the Swiss National Science Foundation (grant no. 200021_184634).

**Author contributions** Q.Z. and Z.S. designed the research and wrote the manuscript. Z.S. performed the analysis. Y.P., D.F., V.P.S., C.-Y.X., W.W. and G.W. discussed and modified the manuscript. Q.Z. and Z.S. contributed equally to the work as the co-first authors and shared equal responsibilities as the corresponding authors.

**Competing interests** The authors declare no competing interests.

**Additional information**
**Correspondence and requests for materials** should be addressed to Qiang Zhang or Zexi Shen.

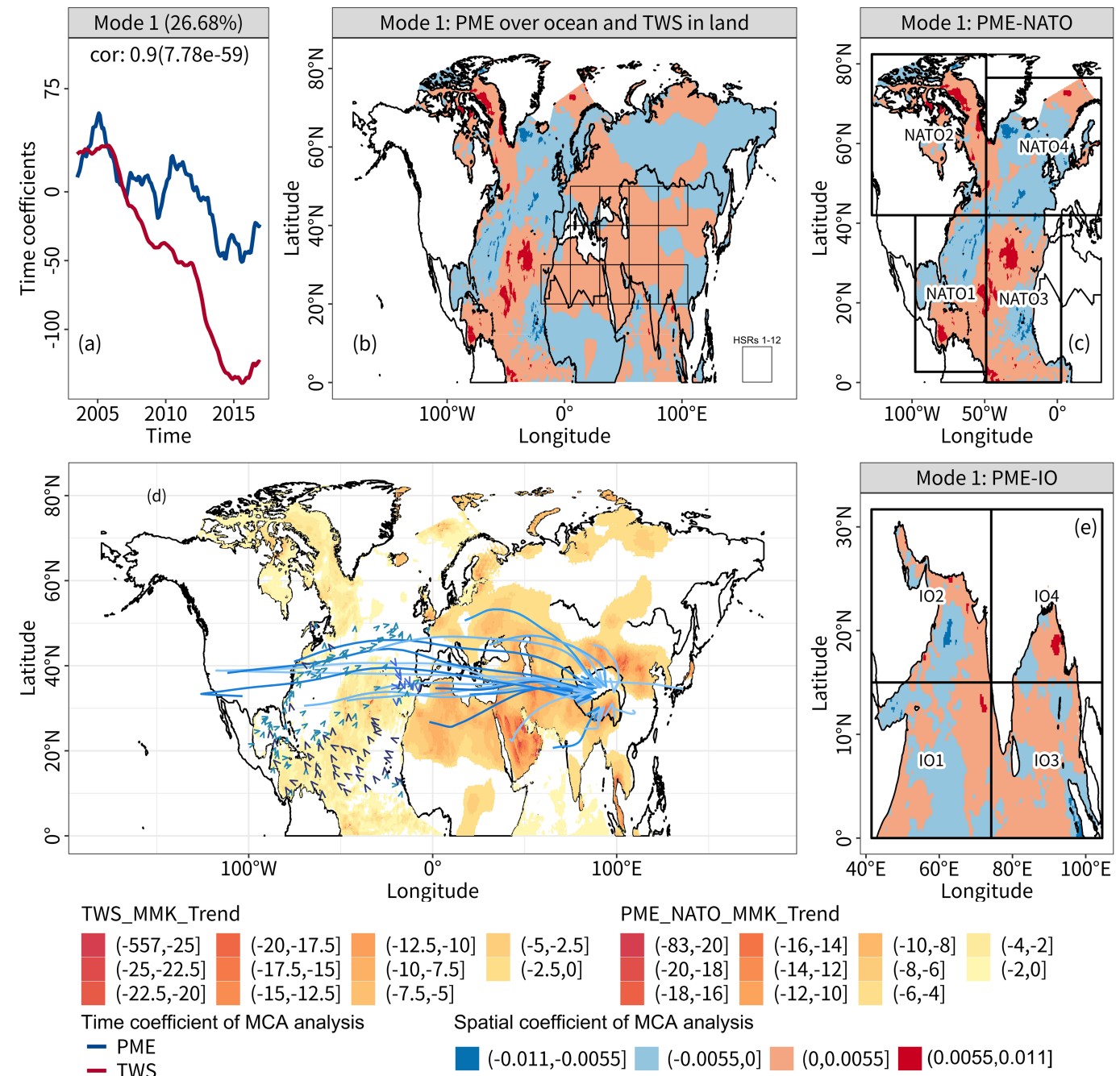

**Extended Data Fig. 1 | Maximum covariance analysis between oceanic PME and TWS in lands. a**, Denotes time coefficients of PME over ocean and TWS across Eurasia under the first leading mode with the explained variance ratio as 26.68%. 'cor.' quantifies the relation between time coefficients PME over oceans and TWS across Eurasia with the *P*-value of 7.78 × 10⁻⁵⁹. **b**, Depicts spatial coefficients of PME over oceans and TWS across Eurasia under the first leading

mode. **c**,**e**, Same as panel **b** but for the North Atlantic (**c**) and the Indian Ocean (**e**). **d**, Presents the modified Mann–Kendall trends in oceanic PME and TWS in lands. The continental world map data[34] in panels **b**–**e** and map data of the TP[35] in panel **d** are acquired from public data sources and plotted using R (ref. [36]). The results used to generate the figure are available through Zenodo[37]. MCA, maximum covariance analysis.

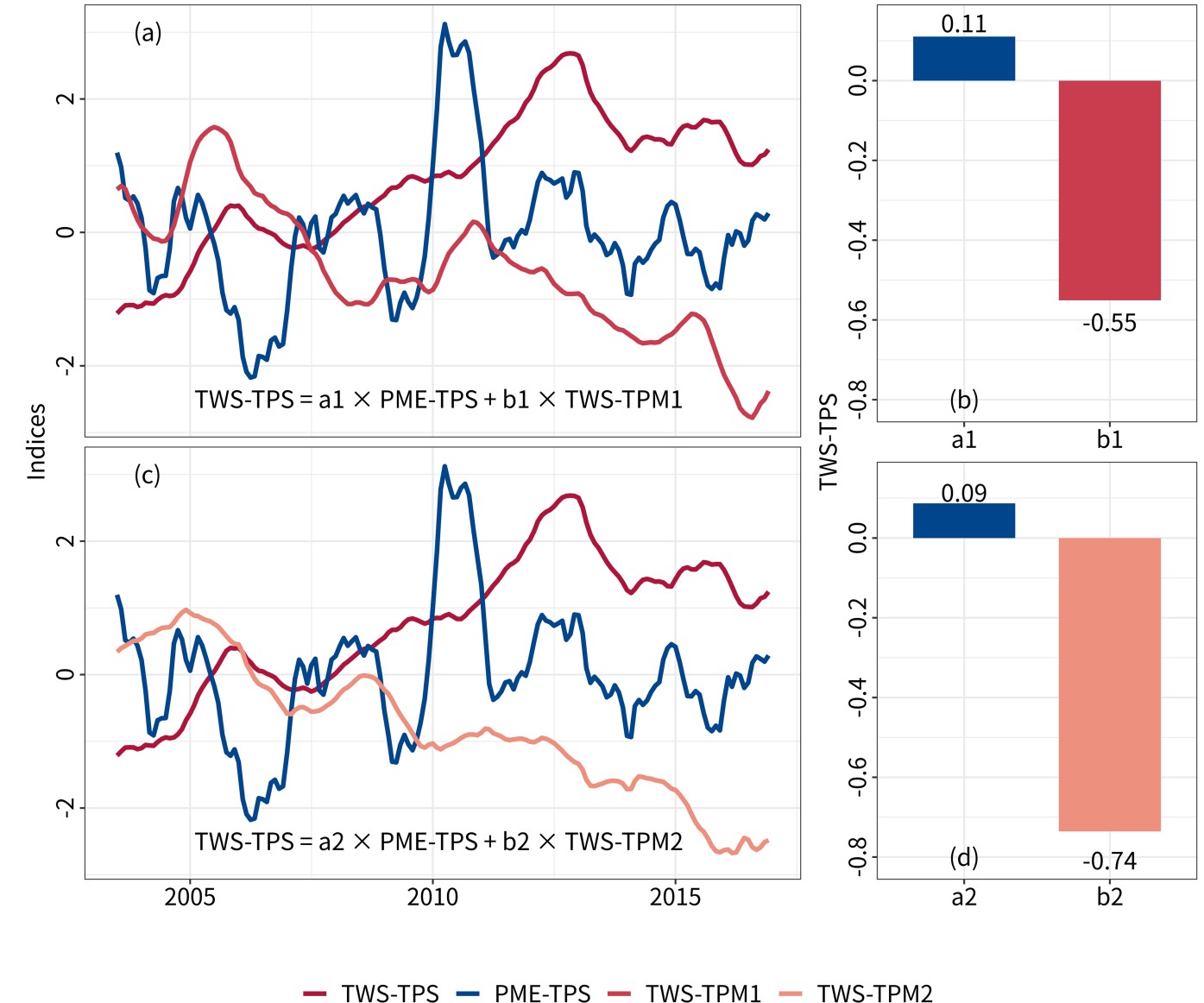

**Extended Data Fig. 2 | Attribution analysis of the TWS in the central TP.**
**a,c**, Temporal variances in TWS and PME in the central TP and TWS in the southwest (TPM1) and southeast (TPM2) TP mountains. **b,d**, Include coefficients *a*1, *b*1, *a*2 and *b*2 of the linear regression models in **a** and **c**. The results used to generate the figure are available through Zenodo[37].

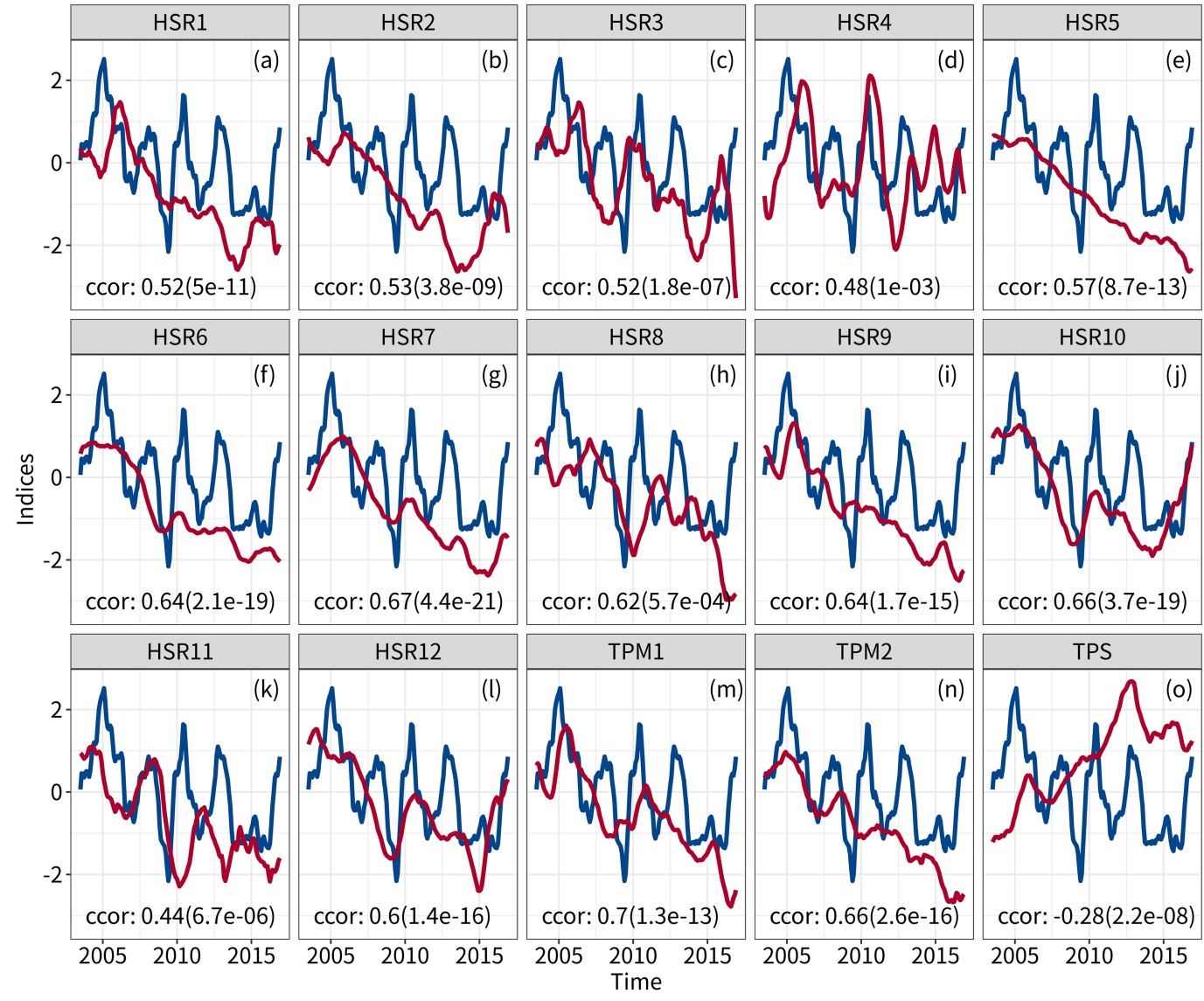

**Extended Data Fig. 3 | PME over NATO3 and continental TWS. a–l**, Comparisons between PME over NATO3 and TWS in HSR1–HSR12. **m,n**, Same as panels **a–l** but for TWS in the southern TP mountains (TPM1 and TPM2). **o**, Same as panels **a–l** but for TWS in the TPS. 'ccor' denotes cross-correlation coefficient and the numbers in the parentheses present *P*-values in the cross-correlation analysis. The results used to generate the figure are available through Zenodo[37].

(a) Mechanism of the interception of the eastward propagation of PME deficit in NATO by TPM 1-2

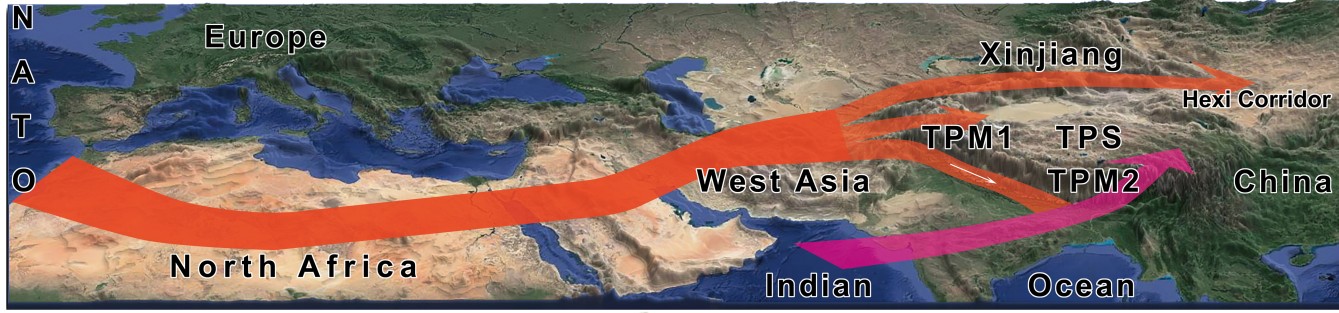

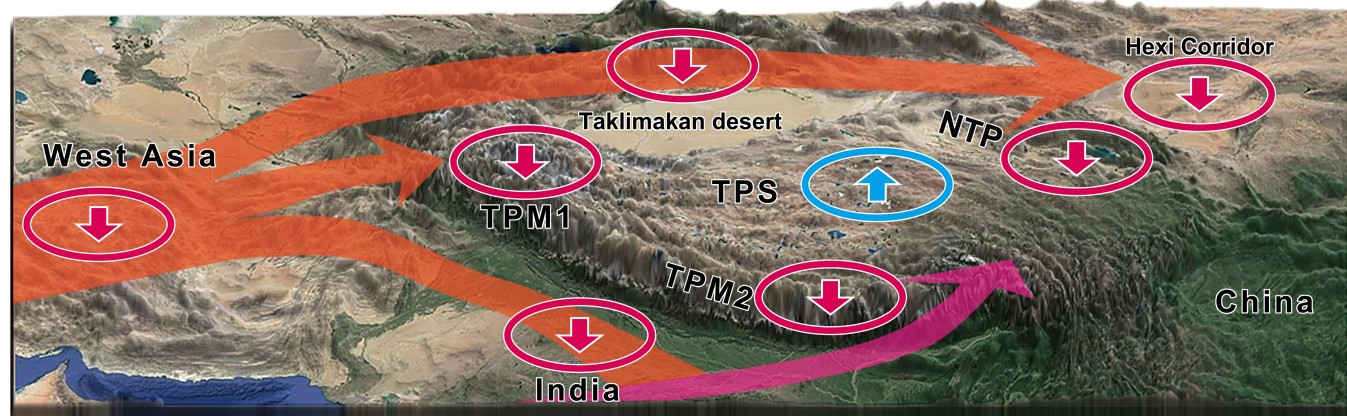

(b) Variation in the interception of the eastward propagation of PME deficit in NATO by TPM 1-2

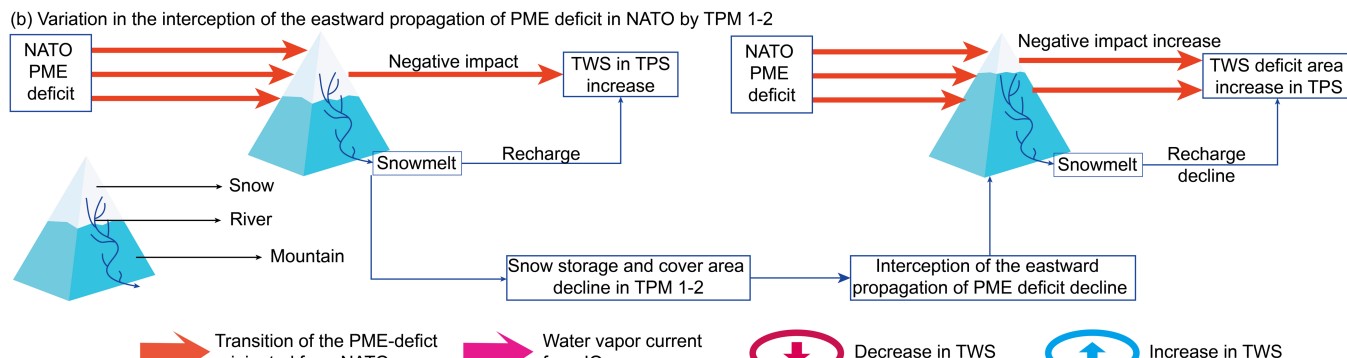

**Extended Data Fig. 4 | Illustration of the blocking effect by HMA.**
**a**, Mechanism of the interception of the eastward propagation of the PME deficit in the North Atlantic by the southern HMA. **b**, Variation in the interception of the eastward propagation of the PME deficit in the North Atlantic by the southern HMA. The southwest (TPM1) and southeast (TPM2) TP mountains are all in the southern HMA. The terrain texture in panel **a** is acquired from Google and the Tangram Heightmapper (https://tangrams.github.io/heightmapper/) and is plotted using the authorized Adobe Photoshop 2021 plug-in named '3D Map Generator – Terrain'. Panel **b** is plotted using the authorized Adobe Illustrator 2021. This figure is based on the results of this study[37].

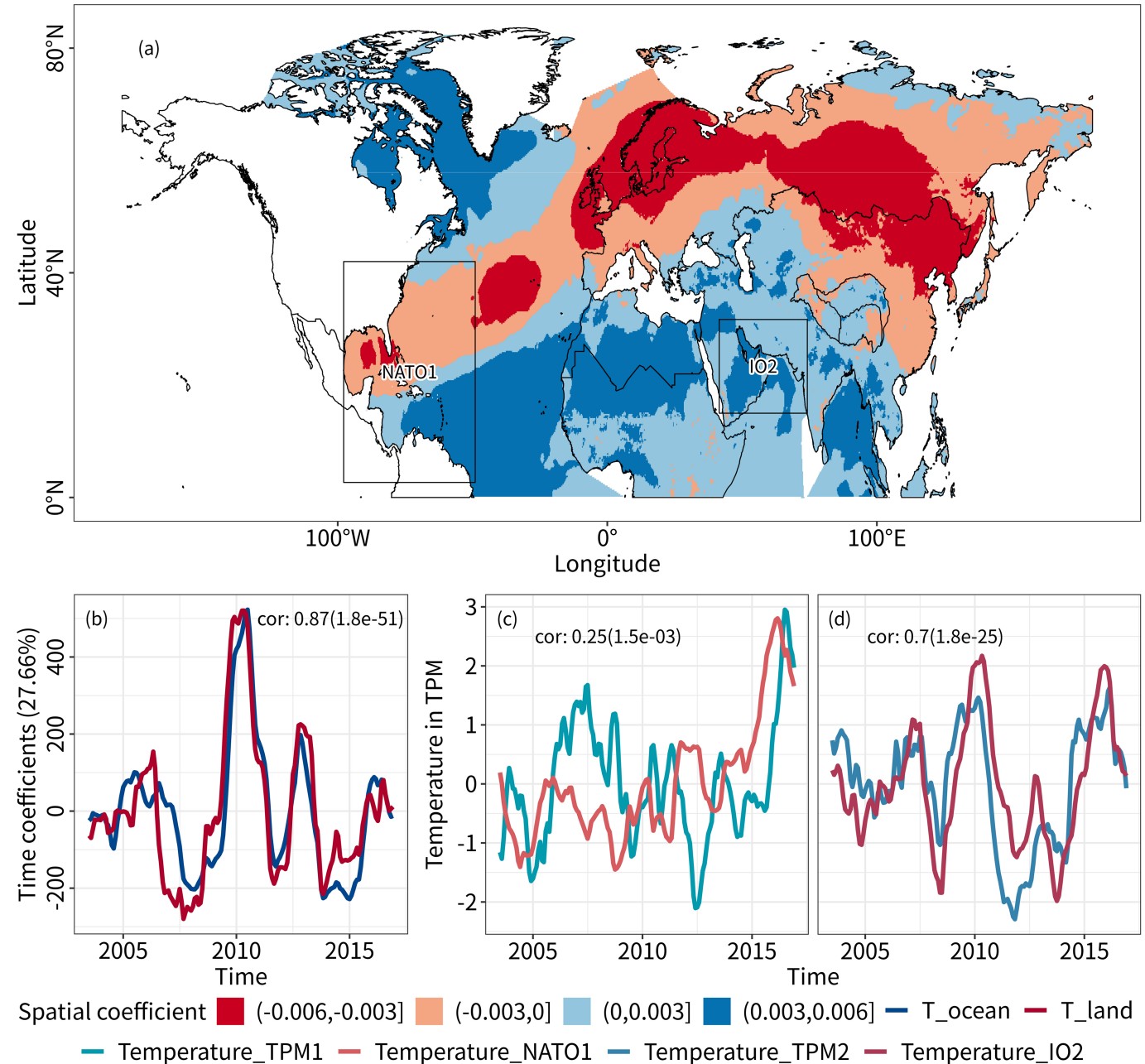

**Extended Data Fig. 5 | Maximum covariance analysis between oceanic and continental T. a**, The spatial coefficients of the air temperature (T) over oceans and that across Eurasia under the first leading mode. **b**, The time coefficients of the air temperature over oceans and that across the Eurasia under the first leading mode with the explained variance ratio as 27.66%. **c**, Temporal variation in air temperature over the southwest North Atlantic (NATO1) and air temperature in the southwest TP mountains (TPM1). **d**, Temporal variation in air temperature in the northwest Indian Ocean (IO2) and air temperature in the southeast TP mountains (TPM2). The continental world map data[34] and map data of the TP[35] in panel **a** are acquired from public data sources and plotted using R (ref. [36]). The results used to generate the figure are available through Zenodo[37].

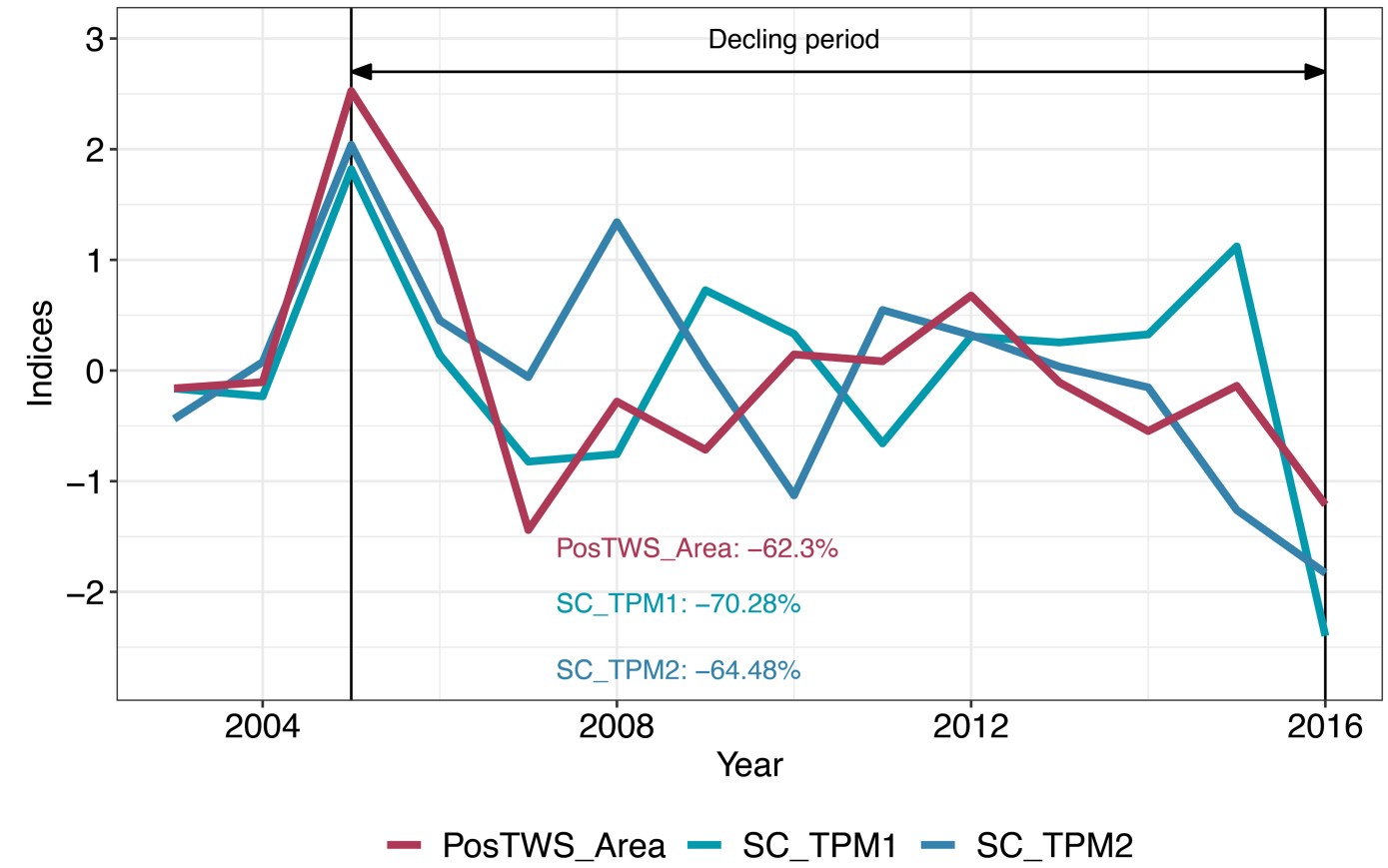

**Extended Data Fig. 6 | Standardized annual snow-cover area and the area under positive annual TWS in the TP.** The PosTWS_Area denotes the standardized area under positive annual TWS in the TP during 2003–2016. The results used to generate the figure are available through Zenodo[37].

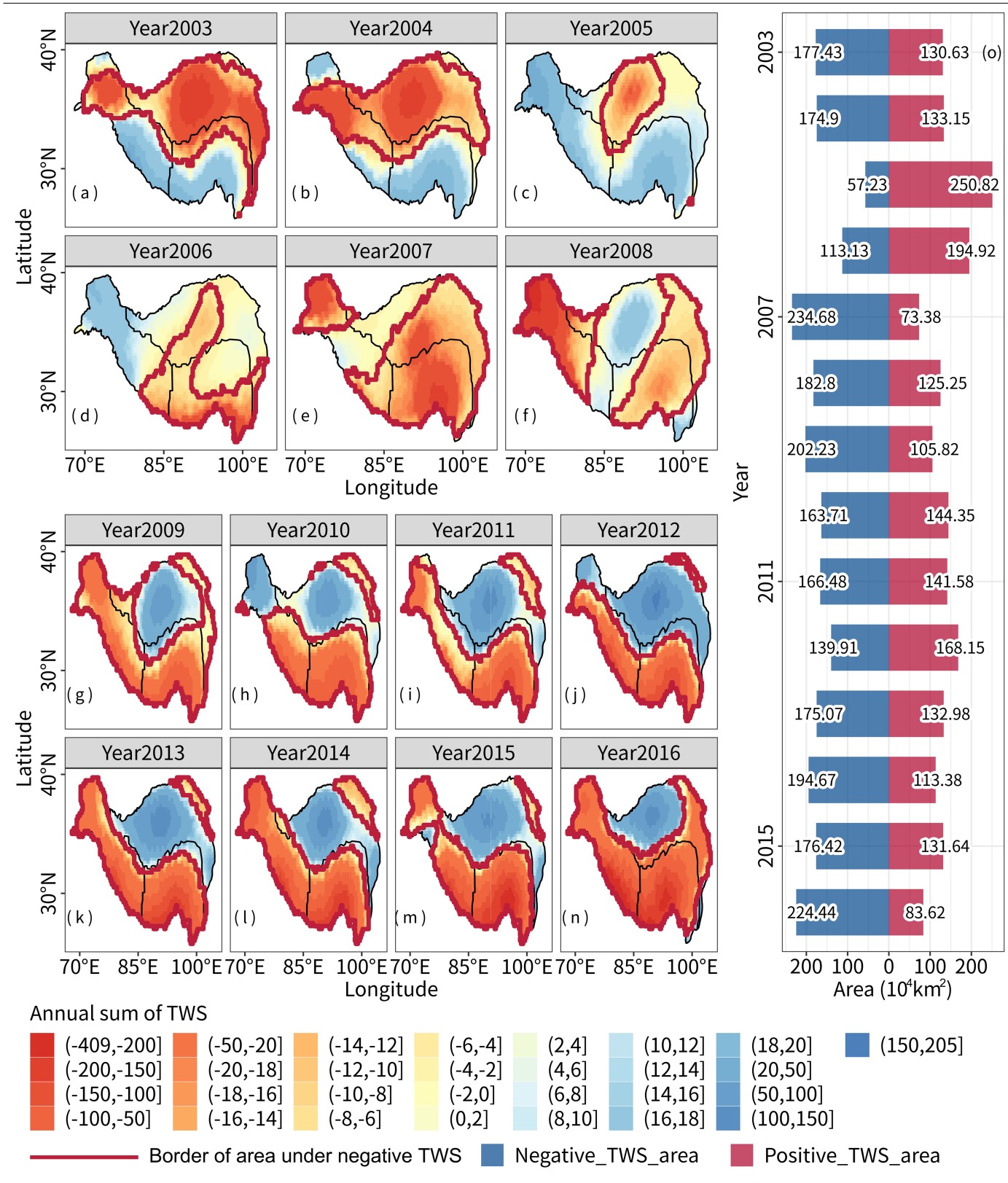

**Extended Data Fig. 7 | Evolution in the area under negative annual TWS in the TP. a–n**, Spatiotemporal transition pattern of the border of the area affected by negative annual sum of TWS in the TP during 2003–2016. **o**, The variation in the area in the TP affected by positive and negative annual sum of TWS. The map data of the TP[35] in panels **a–n** are acquired from public data sources and plotted using R (ref. [36]). The results used to generate the figure are available through Zenodo[37].

## (a) Average TWS non-deficit and deficit area

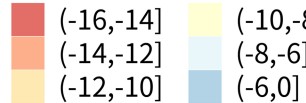

Area (10⁴km²) — $Area\ (10^4 km^2)$

**2020-2051**
- 158.7 (51.5)%
- 166.7 (54.1)%
- 149.4 (48.5)%
- 141.4 (45.9)%

**2051-2082**
- 101.5 (32.9)%
- 106.6 (34.6)%
- 206.5 (67)%
- 201.5 (65.4)%

**2082-2098**
- 82.4 (26.7)%
- 56.4 (18.3)%
- 225.6 (73.2)%
- 251.6 (81.7)%

■ TWS deficit area under SSP245    ■ TWS deficit area under SSP585
■ TWS non-deficit area under SSP245    ■ TWS non-deficit area under SSP585

### (b) SSP245 2020-2051
Max:192.5(62.6%)
Min:101.3(32.8%)

### (c) SSP245 2051-2082
Max:256.8(83.4%)
Min:149.8(48.7%)

### (d) SSP245 2082-2098
Max:258.5(83.7%)
Min:191.2(62%)

### (e) SSP585 2020-2051
Max:171.5(55.8%)
Min:90.5(29.5%)

### (f) SSP585 2051-2082
Max:272.6(88.6%)
Min:156(50.6%)

### (g) SSP585 2082-2098
Max:298.6(97%)
Min:212.6(69.1%)

Latitude

Longitude

**Annual sum of monthly mean TWS**

| | |
|---|---|
| ■ (-16,-14] | ■ (-10,-8] |
| ■ (-14,-12] | ■ (-8,-6] |
| ■ (-12,-10] | ■ (-6,0] |

**Projected northern border of TWS deficit area**
— Border of the minimum TWS deficit area
— Border of the mean TWS deficit area
— Border of the maximum TWS deficit area

**Extended Data Fig. 8 | Future evolution in the area under negative annual TWS in the TP. a**, The average area under annual positive (non-deficit) and negative (deficit) TWS in the TP under SSP245 and SSP585 scenarios, respectively. **b–g**, Projected borders for the maximum, mean and minimum areas under negative TWS in the TP under SSP245 scenario (**b–d**) and SSP585 scenario (**e–g**), respectively. The map data of the TP[35] in panels **b–g** are acquired from public data sources and plotted using R (ref. [36]). The background terrain map in panels **b–g** is from Google Maps acquired using ggmap[38] in R. The results used to generate the figure are available through Zenodo[37].

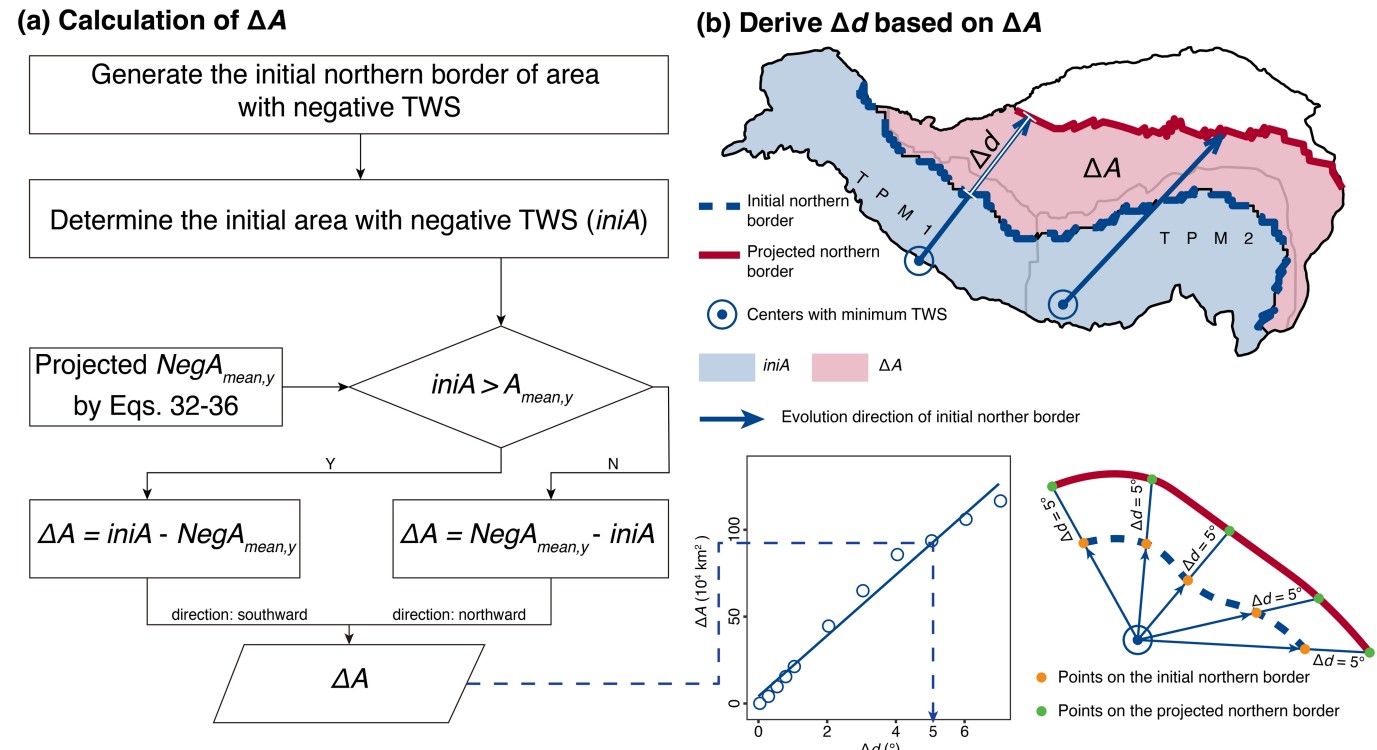

**(a) Calculation of ΔA**

Generate the initial northern border of area with negative TWS

Determine the initial area with negative TWS (*iniA*)

Projected $NegA_{mean,y}$ by Eqs. 32-36

$iniA > A_{mean,y}$

Y  N

$\Delta A = iniA - NegA_{mean,y}$

$\Delta A = NegA_{mean,y} - iniA$

direction: southward  direction: northward

$\Delta A$

**(b) Derive Δd based on ΔA**

Δd

ΔA

T P M 1

T P M 2

- - - Initial northern border

—— Projected northern border

◉ Centers with minimum TWS

iniA ΔA

→ Evolution direction of initial norther border

$\Delta A$ (10⁴ km²)

$\Delta d$ (°)

Δd = 5°

● Points on the initial northern border

● Points on the projected northern border

**Extended Data Fig. 9 | Illustration for the projection of borders in the future.** Determination of the projected northward expansion or southward shrinkage area $\Delta A$ (**a**) and movement distance $\Delta d$ (**b**). $NegA_{mean,y}$ refers to the averaged projected TWS deficit area by four data inputs in year $y$ (see Methods). For the definition of the initial northern border of the TWS deficit area, see Methods. The map of the TP[35] in panel **b** is acquired from a public data source and plotted using R (ref. [36]). The results used to generate the figure are available through Zenodo[37].