## [Peer Review File · Nature]

Manuscript Title: Changes in oceanic climate threaten the sustainability of Asia's water tower

Reviewer Comments & Author Rebuttals

Reviewer Reports on the Initial Version:

Referees' comments:

Referee #1 (Remarks to the Author):

Dear authors, I have reviewed your manuscript entitled "Sustainability of Asian water tower threatened by drying and warming ocean".

The results are of interest to the scientific community but also as information for taking decisions for the benefit of the economy and the population. The authors found interesting and unexpected results, such as the relative importance of the North Atlantic over the Indian Ocean as a moisture source for the TP. However, there are some details I would like to share with you. First, improve the introduction of the manuscript in order to make the objective of the study clearer. Indeed, it is not clear, at least for me, when the authors focus on the relationships between PME and TWS over 3 subregions (SR1, SR2 and SR3) and the contribution from oceanic sources. The title of the study is then, in my opinion, not consequent with these analyses, because the study should be focused on the role of the sources on the precipitation and TWS in the TP. I would appreciate your opinion about it, because to be published in this journal, the study must be concise and focused on the objective of the study without rambling on collateral topics. In this line, I also expected to find the analysis of the optimum lag relationship between the PME and TWS, which was not assessed, or an explanation about it. And I mention this because TWS may be influenced in the medium and long term by the moisture supply from the external sources, taking into account that TWS includes surface but also underground freshwater.

The authors confirmed the crucial importance of Asia as moisture source for the TP. But along the paper there is no explanation about the possible role of recycling in this region, particularly the recycling of precipitation attributed to the moisture transport from the Indian Ocean, and the relationship of the moisture export from Asia to the TP, and the moisture contribution from the IO to Asia. This chain could increase the importance given to the IO in the study, and must be discussed.

In the title of the manuscript you mention: *Ocean warming*. I did not find in the manuscript the rate of warming of the oceanic moisture sources (under the historical and future scenarios)? I

In my opinion, the Supplementary Information is extensive, with multiple figures that make it difficult to follow, you can consider removing some of the Supplementary Figures that can be unnecessary.

The approach is relatively novel and has been previously used for similar studies.

Considering the above comments, I suggest major revisions must be done to be acceptable for publication. I invite the authors to revise their manuscript to address the concerns (some listed below) before a final decision by the Editorial is reached. I strongly ask you to clarify some aspects of the methodology, see the comments below.

Specific comments

1. Lines 26-27. You mean 'a' Lagrangian Partical Dispersion Model and not 'the'.
2. In Figure 1a, is confusing the almost similar colors (yellow-reddish) for some of the arrows that you use to denote the backward direction of air parcels from the regions (e.g AS, EU, etc.)
3. I have a question about Figure 1. Normally, when on route changes are considered, the importance of the sources decrease, decreasing as well the importance of longer regions as moisture sources for the target region. But here you find the opposite. How can you explain it?
4. Lines 135-137: *135 The TWS changes along the trajectories followed by water vapor transported towards the TP are influenced more prominently by the PME deficits in the North Atlantic than by both PME variations in the Indian Ocean and local PME variations.*

According to this sentence, you assessed the TWS over the continents along the trajectories, but, you are following the air parcels initially residing over the TP and not from the rest of Eurasia, so, the relationship between TWS and PME outside the TP deserves further analysis. Just explain it to me and improve the explanation in the manuscript if necessary.

0. Lines 182-183. *Drying over the North Atlantic induced a TWS decrease between mid-latitude Eurasia (20°N-50°N) and the southern TP mountains, but not in the central TP.*

Regarding this sentence, I can understand the analysis for the TP taking into account that backward analysis was performed from this region, but for the rest of Eurasia, how can you explain this?

1. Line 207. Where is Figure 5?
2. Line 252. You don't provide an explanation in terms of the atmospheric circulation and PME for the year 2013.
3. The analysis of TWS under future scenarios seems very interesting.

4. In the Methods

5. Which version of FLEXPART are you using? You should mention it.
6. I wonder why the authors chose 500 000 particles? Is there any criterion for this? Can you explain what is the relation between the number of particles and the resolution of the input data? I have seen the world atmosphere divided into near 2 million of parcels, but not a half of million for a small region like the TP; just clarify it to me and in the manuscript.
7. Lines 350-351. You argue that according to 39, ... *Given that it approximately takes 10 days for an air particle to travel around Earth.* Reference 39 is about mining, so, from where did you take this metric?

Secondly, I have read and used 10 days according to Numaguti (1999). There are also other studies that assess the residence time of the water vapor in the atmosphere, which is a better criterion than a particle travel (in days) around Earth.

8. Line 357. (*this choice is dictated by the available FLEXPART model version*). This is not clear for me, since at least FLEXPART 10 can be executed using ERA5. Authors must clarify this issue.
9. Why 19 regions in the K-mean analysis? Which criterion did you utilised for determining the optimum number of clusters?
10. In line 346 you describe the backward analysis for determining the moisture sources for the TP region. However, from line 359 to 364 you describe the high relationship between the total water vapor released from the TP and precipitation data from ERA5 and GPCC. So, is not clear for me because in the backward analysis for identifying the moisture sources is calculated the moisture gain by the parcels and the total average on the total column. If you calculate the moisture release in the backward analysis, did you compute the moisture gain? Is not clear for me. Also, did you calculate the moisture release in the backward experiment computing PME within the TP and correlated it with the precipitation from ERA5 and GPCC averaged over the TP?. Another doubt, did you run the (E-P) just in the backward mode?
11. The moisture loses (E-P) <0 is not exactly the precipitation. Is has been named as 'Lagrangian precipitation' in previous studies (e.g. Nieto and Gimeno, 2019; Gimeno et al., 2020). In this study you intend to make a validation between the moisture lose computed with FLEXPART and the precipitation from ERA5 and GPCC, only through Pearson correlations, but it is not correct to name is as 'validation', you assess the relationship. Authors must clarify that moisture release is not the precipitation, and remove the word 'validation'. Indeed, for a real validation a correlation is not enough, as you should know.
12. In section 5 the authors describe the regression analysis performed. The TWS is considered as the total water terrestrial water resources, which consider the surface water bodies, so, I wonder why the authors did not make a lag in time analysis to determine the optimum lag in which TWS is affected by moisture contribution from the sources to precipitation over the TP. Indeed, in Supplementary Figure 9 is observed the low correlations.
13. Line 463, which ERA5 reanalysis data?
14. There is no conclusions ?

Referee #2 (Remarks to the Author):

Review of "Sustainability of Asian water tower threatened by drying and warming ocean", by Zhang, Shen et al.

The authors present a very comprehensive study to possible explanations for the spatial variations in trends of terrestrial water storage on the Tibetan Plateau. They use a combination of the FLEXPART particle dispersion model with reanalysis climate data and GRACE remotely sensed TWS data, and argue that TWS decline in the southern TP can be attributed to a deficit in precipitation minus evapotranspiration from the southeast North Atlantic. They further argue that the high mountain ranges in Asia block the propagation of the deficit further onto the central TP which leads to increasing TWS there. Lastly they demonstrate using CMIP6 GCM data how this blocking could weaken in the future, causing also the central TP to witness future TWS deficits.

To my knowledge this is an original approach, it is mostly well described and components of the analysis thoroughly validated. The findings are of broad interest and provide new knowledge about the possible mechanisms of variability in climate change trends across the TP. In some cases I am not fully convinced about the robustness of intermediate conclusions, which propagate into the further analysis. I believe these need further clarification. These are included in the specific comments below.

Title: The title seems somewhat strange in that it could be read as that the ocean is drying up. Consider rewording.

L84-85: I do not understand the statement 'Since glacier melting dominated the TWS changes in the high mountains instead of elevation'.

Figure 1 a+b: The water vapor trajectories are hard to distinguish. I suggest to try more contrasting colors and/or different line types.

Figure S1: Suggest to change the y-axis for panels b-d to be able to see more details, like also done in other rows in the figure.

L139-140: Looking at Figs S1 and S2, the contributions from North Africa are about equal to the North Atlantic. Therefore, I wonder how the authors exactly come to the conclusion that contributions from the North Atlantic dominate over contributions from North Africa? This conclusion is important as it propagates further in the analysis when focus is going to the North Atlantic region. I would guess that the ocean being a body of water supplies more water vapor than the northern African land mass with the dry Sahara, but I cannot deduct that from Figs S1-S2. In Figs S3-S9 correlations for the North Atlantic and Indian Ocean are shown and North Africa is not further investigated whereas the data in Figs S1 and S2 suggest it should. I am no expert in particle dispersion modeling, so the explanation could be something obvious, but then it requires a bit more explanation for the reader. Also, while Figs S3-S9 indeed show better correlations for North Atlantic compared to Indian Ocean, it seems that Indian Ocean still plays a large role, in particular IO4 (Fig S8, panel c).

L198-201: I find the preceding argumentation and interpretation of Extended Data Figures 3 and 4 for the point that the high mountains block the propagation of the PME deficit hard to follow. Therefore it also does not fully convince me and I am not sure how valid the point is. I suggest to revisit lines 186-201 and refer to particular panels in the referred figures and elaborate explanation for this important part of the paper.

L324-327: Why were these particular CMIP6 models included and others not? I also believe MICRO6 is a typo and should be MIROC6.

L491-492 and Figs S18-19: Comparing timeseries of CMIP6 output directly year-to-year to reanalysis data in my experience never provides good correlation. I believe it is more valid to compare climate characteristics averaged over longer periods, like distributions, degree of variability.

Supplementary Table 1: I suggest to omit this table, the statement in the Methods section is sufficient.

A recent relevant paper that was published in the meantime to consider citing where appropriate:
Li, X., Long, D., Scanlon, B. R., Mann, M. E., Li, X., Tian, F., Sun, Z., & Wang, G. (2022). Climate change threatens terrestrial water storage over the Tibetan Plateau. *Nature Climate Change*.
<https://doi.org/10.1038/s41558-022-01443-0>

Author Rebuttals to Initial Comments:

Response to Reviewer 1

1. Summary comments and reply

Dear authors, I have reviewed your manuscript entitled “Sustainability of Asian water tower threatened by drying and warming ocean”.

Reply: Thank you for your time in reviewing our manuscript. To make it easy to follow, under each comment, we have included appropriate reference to the revised manuscript (e.g., line numbers), wherever appropriate. Quoted text from the manuscript is provided in *italics*.

The results are of interest to the scientific community but also as information for taking decisions for the benefit of the economy and the population. The authors found interesting and unexpected results, such as the relative importance of the North Atlantic over the Indian Ocean as a moisture source for the TP. However, there are some details I would like to share with you. First, improve the introduction of the manuscript in order to make the objective of the study clearer. Indeed, it is not clear, at least for me, when the authors focus on the relationships between PME and TWS over 3 subregions (SR1, SR2 and SR3) and the contribution from oceanic sources. The title of the study is then, in my opinion, not consequent with these analyses, because the study should be focused on the role of the sources on the precipitation and TWS in the TP. I would appreciate your opinion about it, because to be published in this journal, the study must be concise and focused on the objective of the study without rambling on collateral topics.

Reply: Thank you for your positive comments on our work. We improved the interpretation parts of the results in the revised manuscript and further analysis was also added.

The reason why we analyzed “*the relationships between PME and TWS over 3 subregions (SR1, SR2 and SR3) and the contribution from oceanic sources*” is explained as follows.

Firstly, it is mainly because that we want to demonstrate the transition process of the changes in oceanic climate to TP. The TP is the part of the SR1-3 but physically separated from the oceans. And the oceans (e.g., Indian Ocean and North Atlantic) are demonstrated in this revision that recharge the water vapor of air particles over lands (e.g., Asia and Africa) that further transit water vapor into the TP (Supplementary Fig. 1). Thus, the changes in oceanic climates might transit to TP by atmospheric circulations leading to consistent TWS variations on route (e.g., SR1 and SR2). If the transition process truly exists, the consistent relationships between PME over oceans and TWS in SR1-3 (including TP) are expected. In this study, we verified the transition process by combining the backward trace analyses and relation analyses. It suggests that the TWS in SR1-3 and southern TP mountains are all significantly related to the PME over northwest (NATO2) and southeast (NATO3) North Atlantic (Supplementary Figure 4g-i) based on the westerlies and the intermedia current (see Figure 1b).

Secondly, it is because that we wanted to verify whether the TWS in lands are influenced by the disturbance on the local PME induced by changes in oceanic climate. We analyzed the relationships between PME over SR1-3 and PME over oceans and the relations between PME and TWS over SR1-3. It suggests that the TWS in lands are not influenced by this process (Supplementary Figure 4d-f and Supplementary Figure 5) but are influenced by long-term changes in oceanic climates (Supplementary Figure 4g-i, Supplementary Figure 6 and Extended Data Figure 3).

Overall, the analysis for the “*relationships between PME and TWS over 3 subregions (SR1, SR2 and SR3) and the contribution from oceanic sources*” are necessary for the study focusing on the impacts on TWS in TP induced by changes in oceanic climates.

In this line, I also expected to find the analysis of the optimum lag relationship between the PME and TWS, which was not assessed, or an explanation about it. And I mention this because TWS may be influenced in the medium and long term by the moisture supply from the external sources, taking into account that TWS includes surface but also underground freshwater.

Reply: Yes, we agree with you on this comment. And we had performed the cross-correlation analysis between the TWS in 14 detailed subregions and (Fig. 2b-c, Supplementary Fig. 6 and Extended Data Figure 3) across the mid-latitude Eurasia and the PME over the North Atlantic by considering the optimum lags (see Method 5). The optimum lag is defined as the lag when the maximum correlations between the TWS and the PME are detected. The interpretation of the cross-correlation analysis sees Lines 144-147 and Lines 161-164.

Lines 144-147: Revised manuscript

Focusing on regions within 20°N-50°N, considering optimum lags, we show high cross correlations between the PME deficit over the southeast North Atlantic and the TWS changes within 12 out of 14 sub-regions discerned along the water vapor propagation routes towards the TP (Fig. 2a-d, Supplementary Fig. 6).

We also analyzed the cross-correlation between TWS in southern TP mountains and PME over North Atlantic with consideration of the optimum lags (see Lines 161-164).

Lines 161-164

And the cross-correlation analysis suggests the decreased TWS over the southern TP mountains can be attributed primarily to the PME deficit from the southeast North Atlantic with considering optimum lags (Extended Data Figs. 3m-n).

The authors confirmed the crucial importance of Asia as moisture source for the TP. But along the paper there is no explanation about the possible role of recycling in this region, particularly the recycling of precipitation attributed to the moisture transport from the Indian Ocean, and the relationship of the moisture export from Asia to the TP, and the moisture contribution from the IO to Asia. This chain could increase the importance given to the IO in the study, and must be discussed.

Reply: We have performed additional analysis and demonstrated the process of the moisture transport from oceans to TP. Given that the SR3 covers most area of the TP, the additional backward trace analyses were performed for particles initially residing over the SR1 and SR2 in this revision. According to Supplementary Fig. 1, it suggests that the oceans (e.g., Indian Ocean and North Atlantic) recharge the water vapor of air particles over lands (e.g., Asia and Africa) that further transit water vapor into the TP by the westerlies and the intermedia current (Figure 1a-b).

We agree with you on that the IO also plays a key role on impacting the climatic conditions in Asia and TP, and have attached discussions to the revised version of the manuscript (Lines 101-109).

Lines 101-109: Revised discussion

Distance to the TP has negative influence on the contributions from source regions to TP. Even though, it should be noted that the ocean is the primary continental water source region in a large-scale circulation. The oceans (e.g., Indian Ocean and North Atlantic) recharge the water vapor of air particles over lands (e.g., Asia and Africa) that further transit water vapor into the TP (Supplementary Fig. 1) via the two main trajectories. The first is from the North Atlantic to the western TP transported by the westerlies (Figure 1a-b), and the other is the intermedia current (see Figure 1a-b) that brings water vapor from North Atlantic and Indian Ocean into the southern TP.

In the title of the manuscript, you mention: Ocean warming. I did not find in the manuscript the rate of warming of the oceanic moisture sources (under the historical and future scenarios)?

Reply: We agree with you on that the title might cause some misunderstanding. We have now revised the title to “Changes in oceanic climate threaten the sustainability of Asian water tower”. The oceanic moisture sources for TP include North Atlantic and Indian Ocean. And the connections between the oceanic air temperature variations and air temperature variations in TP have been demonstrated (Extended Data Fig. 5). The air temperature in southwest TP mountain is mainly influenced by the air temperature over southern North Atlantic. And the air temperature in southeast TP mountain is connected to the Indian Oceans.

In this revision, the warming rate of the air temperature over oceans is concerned. And we added this information in the revised manuscript:

Lines 241-244 Revised manuscript

Influenced by the warming air temperature by 1.8~3.9 °C over oceans (Supplementary Fig. 10) and climatic drying impact of the southeast North Atlantic, the accelerated melting of glaciers and snow and TWS deficit continuously weaken the blocking effect of HMA during 2020-2098 under SSP245 and SSP585.

In my opinion, the Supplementary Information is extensive, with multiple figures that make it difficult to follow, you can consider removing some of the Supplementary Figures that can be unnecessary.

Reply: We have now removed repetitive Supplementary and Extended Data Figures. By doing so, we reduced the number of Supplementary Figures from 27 to 25 (8 figures

contained in our first submission were removed and 6 new figures were added in response to comments of the reviewers) and the number of Extended Data Figures from 10 to 9. The detailed revision note is as follows (Table 1 in reply).

Table 1 in reply. Reduction of the Supplementary and Extended Data Figures

Original Figures in the first submission	Revision
Supplementary Figures 3-8	Compressed into Supplementary Figure 4
Supplementary Figure 9	Remove the map for NATO1-4, IO1-4 and SR1-3. The revised plot sees Supplementary Figure 5.
Supplementary Figures 11 and 13	Remove the modified Mann-Kendall trends in TWS. The revised plots see Supplementary Figures 7 and 9.
Supplementary Figures 12	Remove the abundant third and fourth columns of the plot. The revised plot sees Supplementary Figures 8
Supplementary Figures 22-25	Compressed into the Supplementary Figure 21.
Extended Data Figure 1	Removed in the revision submission.
Supplementary Table 1	Removed in the revision submission.

The approach is relatively novel and has been previously used for similar studies. Considering the above comments, I suggest major revisions must be done to be acceptable for publication. I invite the authors to revise their manuscript to address the concerns (some listed below) before a final decision by the Editorial is reached. I strongly ask you to clarify some aspects of the methodology, see the comments below.

Reply: Thank you for your positive comments on our work. We have revised the manuscript by following your specific suggestions.

2. Specific comments

Comment 1

Lines 26-27. You mean ‘a’ Lagrangian Partical Dispersion Model and not ‘the’.

Reply: We replaced ‘the’ with ‘a in the revised manuscript.

Lines 29-30 Revised manuscript

...using a Lagrangian Particle Dispersion Model and satellite observations from 2003-2016 ...

Comment 2

In Figure 1a, is confusing the almost similar colors (yellow-reddish) for some of the arrows that you use to denote the backward direction of air parcels from the regions (e.g AS, EU, etc.)

Reply: Thank you for your suggestions. We also apply the red-yellow colors for trajectories to distinguish them from background information in Figure 1. But, in this revision, different line shapes are applied for the trajectories. Besides, we also attached a Supplementary Figure 25 displaying clustered trajectories one by one for a better illustrative purpose. In the Supplementary Figure 25, we have listed 20 trajectories instead of 19 trajectories. The reason for that is answered in following reply to Comment 13.

Comment 3

I have a question about Figure 1. Normally, when on route changes are considered, the importance of the sources decreases, decreasing as well the importance of longer regions as moisture sources for the target region. But here you find the opposite. How can you explain it?

Reply: For your first puzzle that “*when on route changes are considered, the importance of the sources decreases*”, we agree with you on this comment. When the on-route changes are considered instead of being disregarded, particles should meet the condition mentioned in the Equations 3-4. And when the on-route changes are disregarded, all particles should be included into the calculation of the relative contributions. However, in the first submission, we confused these two methods. In this revision, we corrected this confusion (see Lines 95-101 and Lines 491-515). In this sense, when on route changes are considered, the importance of the source decreases (Figure 1c-e). Thanks for your insightful suggestion here.

For your second puzzle that “*decreasing as well the importance of longer regions as moisture sources for the target region*”, we also agree with you on this point. In this revision, we added the discussion about the impact on the relative contribution induced by the distance (Lines 101-109). And when the on-route changes are considered, the relative contributions from Asia, Indian Ocean, Africa and North Atlantic Ocean ranks from the largest to the smallest while their distances to TP ranks from the smallest to the largest (Figure 1c). This result complies with the aforementioned statement on the impacts of distance. However, when the on-route changes are disregarded, we find that the contribution from the North Atlantic is larger than the one from the Indian Ocean. This is because that the number of particles from North Atlantic is far larger than the number of particles from Indian Ocean disregarding on-route changes, which can be demonstrated by the number of clustered trajectories (Figure 1b). The more particles, the more vapor brought into TP. Thus, the North Atlantic contributes more than Indian Ocean when the on-route changes are disregarded.

Lines 95-101

When accounting for on-route changes (see Methods), we find long-term average RCs of ~42% from Asia, ~7% from the Indian Ocean, ~4% from North Africa, and ~3% from the North Atlantic (Fig.

1c). Disregarding on-route changes does not dramatically alter the outcomes (Fig. 1c). Crucially, during 98% of the study period (77% when disregarding on-route changes), the above four source regions account for over 75% of the total water vapor input to the TP (Fig. 1d-e).

Lines 491-515

Lines 491-493

When accounting for water vapor variations (i.e., recharge or loss) along the transitional routes, the considered air particles show positive variations in water vapor inside the source region (Equation 4).

Lines 495-497

Meanwhile, the considered air particles also should be positive variations in water vapor before reaching the TP when accounting for on-route changes (Equation 5).

Lines 512-515

When accounting for the on-route variation in water vapor, k_i is the number of selected particles (see Equations 3-5) from the i_{th} source regions. When disregarding water vapor on-route variations, k_i refers to the number of all particles from the i_{th} source region.

Comment 4

Lines 135-137: 135 The TWS changes along the trajectories followed by water vapor transported towards the TP are influenced more prominently by the PME deficits in the North Atlantic than by both PME variations in the Indian Ocean and local PME variations. According to this sentence, you assessed the TWS over the continents along the trajectories, but, you are following the air parcels initially residing over the TP and

not from the rest of Eurasia, so, the relationship between TWS and PME outside the TP deserves further analysis. Just explain it to me and improve the explanation in the manuscript if necessary.

Reply: We agree with you on this comment, and we analyzed the relations between TWS and PME over the 3 subregions which are outside the TP in Lines 121-136. Besides, we also further dispersed the 3 subregions as 14 small-scale subregions, and analyzed the cross correlations between the TWS in detailed 14 sub-regions and PME over North Atlantic with considering optimum lags (see Lines 144-147).

In addition, by following your suggestion, we also performed the backward trace analyses for particles initially residing over the SR1 and SR2 and added discussion about it in this revision (see Lines 103-106). It demonstrates that the oceans (e.g., Indian Ocean and North Atlantic) recharge the water vapor of air particles over the lands (e.g., Asia and Africa) that further transit water vapor into the TP by the two main trajectories. (Supplementary Fig. 1). The reason why we did not perform backward trace analysis for SR3 is that the SR3 covers most area of the TP for which backward trace analysis has already been done.

Lines 103-106

The oceans (e.g., Indian Ocean and North Atlantic) recharge the water vapor of air particles over the lands (e.g., Asia and Africa) that further transit water vapor into the TP (Supplementary Fig. 1) via the two main trajectories.

Lines 121-136

Seasonal shifts within the source regions complicate the pattern of PME along the routes. Poor correlations are detected between (i) the PME across three large-scale sub-regions (SR1-3, see Supplementary Fig. 4a-c) along the trajectories towards the

TP and (ii) PME in regions of the North Atlantic and Indian Ocean (Supplementary Figs. 4d-f). The TWS changes, however, can be attributed to the PME deficit transmitted from the water vapor source regions to the TP⁷. The TWS in SR1-3, for example, is significantly related to the PME deficit over the northwest and southeast North Atlantic (correlation coefficients of 0.47 to 0.60, $p < 0.01$) (Supplementary Figs. 4g-i), In contrast, TWS variations in SR1-3 are poorly correlated with PME over regions of the Indian Ocean and local PME variations (correlation coefficients between -0.07 to 0.40) (Supplementary Figs. 4g-i and Supplementary Fig. 5). In addition, relations between TWS over SR1-3 and PME over Indian Ocean are significantly declining with the distance closer to TP (Supplementary Figs. 4g-i). Instead, PME over northwest and southeast North Atlantic have persistently significant relations with TWS over SR1-3. Thus, further analysis mainly focuses on the impact on TWS over the TP induced by PME deficit over North Atlantic.

Lines 144-147

Focusing on regions within 20°N-50°N, considering optimum lags, we show high cross correlations between the PME deficit over the southeast North Atlantic and the TWS changes within 12 out of 14 sub-regions discerned along the water vapor propagation routes towards the TP (Fig. 2a-d, Supplementary Fig. 6).

Comment 5

Lines 182-183. Drying over the North Atlantic induced a TWS decrease between mid-latitude Eurasia (20°N-50°N) and the southern TP mountains, but not in the central TP. Regarding this sentence, I can understand the analysis for the TP taking into account that backward analysis was performed from this region, but for the rest of Eurasia, how can you explain this?

Reply: By following your suggestion, we have performed additional backward trace analysis for particles initially residing over the SR1 and SR2. Since the SR3 covers

most area of the TP, thus, we did not perform additional backward trace analysis for the SR3. Additional backward trace analysis demonstrates that the oceans (e.g., Indian Ocean and North Atlantic) recharge the water vapor of air particles over lands (e.g., Asia and Africa) that further transit water vapor into the TP (Supplementary Fig. 1) by the two main trajectories. The first is from the North Atlantic to the western TP transported by the westerlies (Figure 1a-b), and the other is the intermedia current (see Figure 1a-b) that brings water vapor from North Atlantic and Indian Ocean into the southern TP.

In addition, based on the cross-correlation analyses (Supplementary Fig. 6 and Extended Data Fig. 3), it suggests that the PME deficit over the southeast North Atlantic mainly transited by westerlies leads to coherent declines in TWS across 14 subregions across the mid-latitude Eurasia and the southern TP mountains but not in TWS over the central TP.

Comment 6

Line 207. Where is Figure 5?

Reply: This was a mistake from our side: we meant to refer to Extended Data Figure 5 (now is Extended Data Figure 4). We corrected the corresponding sentence as follows:

Lines 184-186 in revised manuscript

According to our FLEXPART simulations, the propagation routes of the PME deficit from the southeast North Atlantic towards the central TP are blocked by the HMA's high elevations and split into three trajectories (Fig. 1-3 and Extended Data Fig. 4a).

Comment 7

Line 252. You don't provide an explanation in terms of the atmospheric circulation and PME for the year 2013.

Reply: Thanks for the comment. We performed the atmospheric circulation analysis for

the year 2013, and compared it with long-term clustered trajectories during 2003-2017. As the following figure (Figure 1 in reply) suggests, trajectories for year 2013 have no significant difference. Thus, we only keep the long-term trajectories for the illustration of atmospheric circulation in the revised manuscript (Figure 1a-b).

Figure 1 in reply: Comparison between the clustered trajectories for year 2013 and the period of 2003-2017.

Besides, since the direct cause for the abrupt TWS changes in 2013 is attributed to a persistent northward expansion of the TWS deficit in TP since 2009. We have reworded the description in Lines 230-235.

Lines 230-235 Revised interpretation:

Previous studies have shown that the TWS started to decrease over the central TP in 2013^{5,30}; however, the cause of such abrupt TWS decline has not been sufficiently explained. The persistent northward expansion of the TWS deficit in TP since 2009 mainly driven by westerlies is attributed as the direct cause for the abrupt decrease in 2013 (Figs. 2-3 and Extended Data Fig. 6-7).

Comment 8

The analysis of TWS under future scenarios seems very interesting.

Reply: Thanks for your encouraging comments!

Comment 9

Which version of FLEXPART are you using? You should mention it.

Reply: Thanks for your note. We are using FLEXPARTv10.4 in this study. And we have noted this information in the revised manuscript.

Comment 10

I wonder why the authors chose 500 000 particles? Is there any criterion for this? Can you explain what is the relation between the number of particles and the resolution of the input data? I have seen the world atmosphere divided into near 2 millions of parcels, but not a half of million for a small region like the TP; just clarify it to me and in the manuscript.

Reply: According to the guideline of the FLEXPARTv10.4 model³⁹, there is no relation between the number of particles and the resolution of the input data³⁹. On the other hand, the resolution of the input data depends on the data source³⁹. Besides, the guideline points out that the choice of the number of particles depends on the scale of

the target region and computation resource³⁹. For example, the guideline suggests 900,000 particles for a small-scale simulation³⁹. Besides, another regional study also applies thousands of particles for the determination of the moisture sources for the Western North American Monsoon⁴⁷.

We have added a discussion regarding this in the revised manuscript (Lines 457-458).

Lines 457-458

The number of the particles depends on the scale of the target region and computation resources^{39,47}.

Comment11

Lines 350-351. You argue that according to 39, ... Given that it approximately takes 10 days for an air particle to travel around Earth. Reference 39 is about mining, so, from where did you take this metric? Secondly, I have read and used 10 days according to Numaguti (1999). There are also other studies that assess the residence time of the water vapor in the atmosphere, which is a better criterion than a particle travel (in days) around Earth.

Reply: Thank you for your insightful review. It is a kind of typo mistake. The original reference 39 is an example illustrating the bilinear interpolation of the GRACE data. Following your suggestion, we improved the interpretation of the method by citing more appropriate reference here in the revised manuscript.

Lines 460-461 Revised interpretation

Given that the residence time of the water vapor in the atmosphere is approximate 10 days^{48,49}, we perform backward simulations from day 1 to day 10 of each month.

Original reference 39

39. Shen, Z., Zhang, Q., Piao, S., Peñuelas, J., Stenseth, N. C., Chen, D., et al. (2021). Mining can exacerbate global degradation of dryland. *Geophysical Research Letters*, 48, e2021GL094490. <https://doi.org/10.1029/2021GL094490>

Comment 12

Line 357. (this choice is dictated by the available FLEXPART model version). This is not clear for me, since at least FLEXPART 10 can be executed using ERA5. Authors must clarify this issue.

Reply: The reason why we applied ERA-interim is that the ERA5 data supporting the run of FLEXPARTv10.4 is not accessible for the public users yet. We clarified this in the revised manuscript.

Lines 469-471 in revised interpretation

this choice is due to the fact that ERA5 data supporting the run of FLEXPARTv10.4 is not accessible to the public user yet, details see https://www.flexpart.eu/flex_extract/Ecmwf/access.html

Comment 13

Why 19 regions in the K-mean analysis? Which criterion did you utilized for determining the optimum number of clusters?

Reply: Based on the determination standard⁵⁶ for the optimal number of the clusters using k-means method, we initially determined the optimal number of the clusters for the trajectories is 3, and we further derived the 3 clusters of the trajectories from all trajectories (Supplementary Fig. 24). It suggests that there are only 3 eastward trajectories into TP without denoting the trajectory from the Indian Ocean

(Supplementary Fig. 24). Although 3 is the optimal number of clusters, the result could not illustrate enough information about the transition path from the source regions to the TP. As for the criterion for determining the optimum number of clustered trajectories, we do not find relative standards in the handbook of the FLEXPARTv10.4 model and we also find various number of clusters used in other studies^{47,51}. Thus, the optimum number of clusters is generally determined on a case-by-case basis.

In this study, we attempt to enrich the information of the trajectories and avoid far overlapping of the trajectories at the same time. And we set the number of the clusters as 20, and derive 20 clustered trajectories (Supplementary Fig. 25). However, the 9th trajectory has an abnormal turn from East Asia to TP, attributed to the disturbance induced by the zero-longitude centered map projection on the process of clustering trajectories. Since the rest of 19 trajectories are enough for illustrating the transition path from source regions to TP, thus, we finally only keep the 19 clustered trajectories in the main text. In this revision, we added the detailed 20 trajectories in the Supplementary Fig. 25 for readers information and state the reason why we keep the 19 trajectories (Methods and Supplementary Note 1).

Comment 14

In line 346 you describe the backward analysis for determining the moisture sources for the TP region. However, from line 359 to 364 you describe the high relationship between the total water vapor released from the TP and precipitation data from ERA5 and GPCC. So, it is not clear for me because in the backward analysis for identifying the moisture sources is calculated the moisture gain by the parcels and the total average on the total column. If you calculate the moisture release in the backward analysis, did you compute the moisture gain? It is not clear for me. Also, did you calculate the moisture release in the backward experiment computing PME within the TP and correlated it with the precipitation from ERA5 and GPCC averaged over the TP? Another doubt, did you run the (E-P) just in the backward mode?

Reply: In response to your first question, in the Equation 4, we have calculated the net moisture gain over source regions in the backward simulation and applied the net moisture gain >0 as the selection rule for particles. However, we did not directly determine the water vapor relative contribution from source regions by dividing the total vapor release over the TP with the moisture gain over the source regions. This is because that total moisture gain from all source regions is far larger than the total release over the TP (Supplementary Fig. 12).

In this study, to determine the relative contributions from source regions, we firstly calculated the moisture gain over source regions in Equation 4 and moisture variations on the route to TP in Equation 3. And finally, with Equation 4 and Equation 5 as the conditions, we selected particles which meet the demand that the net moisture gain before reaching TP is still positive. Thirdly, we calculate the total releases over the TP for selected particles from different source regions in Equation 6, and then determine the relative contributions of water vapor by dividing the total water vapor releases over TP by the release over the TP from every source region in Equation 6. Above is the calculation process when the on-route changes are accounted for. When the on-route changes are disregarded, the calculation process is the same as above but for all particles into TP.

We also calculated the relative contribution based on moisture gain from sources by following your suggestion. Noteworthy is that the relative contributions calculated based on (i) moisture gain from sources (Supplementary Fig. 12) and (ii) moisture release over TP from sources (Figure 1c) are consistent. They both draw a conclusion that the Asia, Indian Ocean, North Africa and North Atlantic are the main water source regions for the TP. In this revision, we compared these two methods and stated the consistency in the results (Lines 517-524).

Lines 517-524

Besides, there are another method⁵¹ that determines the RC by dividing the total water vapor release over the target region by the moisture gains over the source regions. To avoid impact induced by the differences between methods on the results, we also calculate the RC using this method. It suggests that both methods come to the same conclusion that the Asia, Indian Ocean, North Africa and North Atlantic are the four main water vapor source regions for TP (Supplementary Fig. 12 and Figure 1c). However, given the moisture gains over the source regions are far larger than the total water vapor release in the target region, thus we apply the first method in this study.

The second question in concern 14: Also, did you calculate the moisture release in the backward experiment computing PME within the TP and correlated it with the precipitation from ERA5 and GPCC averaged over the TP?

Reply: Thanks for the comments. Yes, we did calculate the moisture release in the backward experiment computing PME within the TP. In this study, we calculate the total water release for all particles residing over the TP when the $(E-P) < 0$. The original description “*The total water vapor released from the TP is calculated by Eq. 2 (in this case, $n=500,000$)*” is inappropriate, which might arouse misunderstandings. Thus, we revised the description in the manuscript accordingly in Lines 473-476.

Above calculation is based on previous study⁵², which demonstrated that “*Eq. 2 can diagnose E-P, but not E or P individually*”. It also stated that “*When rain falls, it normally clearly exceeds evaporation. Therefore, by assuming that E and P cannot coexist in the same location at the same time, instantaneous rates of evaporation $E_i = E - P$ when $E - P > 0$, or precipitation $P_i = P - E$ when $E - P < 0$, can be diagnosed. (where i denotes time i)*”. Thus, we calculated the total moisture release for all particles residing over the TP when the $(E-P) < 0$, and compared it with the GPCC and ERA5

precipitation data. We added relative reference to this description.

Lines 473-476 Revised description

The total water vapor release for particles residing over the TP is calculated by Eq. 2 when $E-P < 0$ ⁵², and is compared to the standardized regional average precipitation obtained from ERA5 and GPCC.

The third question in concern 14: Another doubt, did you run the (E-P) just in the backward mode?

Reply: Yes. According to previous studies^{39,47,51}, both forward- and backward-only simulations can successfully facilitate the determination of the water vapor source regions. In this study, we only run the FLEXPARTv10.4 model for the determination of the moisture source regions for TP, SR1 and SR2 in the backward mode.

Comment 15

The moisture loses $(E-P) < 0$ is not exactly the precipitation. It has been named as ‘Lagrangian precipitation’ in previous studies (e.g. Nieto and Gimeno, 2019; Gimeno et al., 2020). In this study you intend to make a validation between the moisture loss computed with FLEXPART and the precipitation from ERA5 and GPCC, only through Pearson correlations, but it is not correct to name it as ‘validation’, you assess the relationship. Authors must clarify that moisture release is not the precipitation, and remove the word ‘validation’. Indeed, for a real validation a correlation is not enough, as you should know.

Reply: We agree with you on this comment. The moisture losses $(E-P) < 0$ over an area is not exactly the precipitation, but has a positive effect on the formation of the precipitation^{39,47,51} over the area. Thus, the positive relation between them could be a tool to evaluate the relative accuracy of the simulation results. If the relation is negative, the modelling results might be inaccurate. In our case, the total released water vapor

over TP is highly related to both GPCC and ERA5 precipitation, with correlation coefficients between 0.85-0.86, which demonstrates the reliability of the simulated results here.

By firmly following your instruction, we have clarified that the moisture release is not the precipitation and removed the word “validation” in the revised manuscript (Lines 471-482).

Lines 471-482

A comparison is performed between the simulated total water vapor release and ERA5 and GPCC precipitation over TP to investigate the sensitivity of simulation results⁵¹. The total water vapor release for particles residing over the TP is calculated by Equation 2 when $E-P < 0$ ⁵², and is compared to the standardized regional average precipitation obtained from ERA5 and GPCC. In both cases, comparisons are performed by employing correlation analysis. It should be noted that the total water vapor release is not equal to precipitation theoretically, but it has the positive effect on the formation of the precipitation^{39,47,51}. The total released water vapor over TP is highly correlated to both GPCC and ERA5 precipitation, with correlation coefficients between 0.85-0.86 ($p.value < 0.01$, see Supplementary Fig. 11). We consider this to be as an acceptable accuracy for the FLEXPART results (Supplementary Fig. 11).

Comment 16

In section 5 the authors describe the regression analysis performed. The TWS is considered as the total water terrestrial water resources, which consider the surface water bodies, so, I wonder why the authors did not make a lag in time analysis to determine the optimum lag in which TWS is affected by moisture contribution from the sources to precipitation over the TP. Indeed, in Supplementary Figure 9 is observed the low correlations.

Reply: Thanks for the suggestion. We performed the cross-correlation analyses between the TWS in 14 detailed subregions (Fig. 2b-c and Supplementary Fig. 6) across the mid-latitude Eurasia and the PME over the North Atlantic with consideration of the optimum lags (see Method 5). The optimum lag is defined as the lag when the maximum correlations between the TWS and the PME are detected. The interpretation of the cross-correlation analysis can be found in Lines 144-147.

Lines 144-147: Revised manuscript

Focusing on regions within 20°N-50°N, considering optimum lags, we show high cross correlations between the PME deficit over the southeast North Atlantic and the TWS changes within 12 out of 14 sub-regions discerned along the water vapor propagation routes towards the TP (Fig. 2a-d, Supplementary Fig. 6).

We also analyzed the cross-correlation between TWS in southern TP mountains and PME over North Atlantic by considering the optimum lags (see Lines 161-164).

Lines 161-164

The cross-correlation analysis suggests the decreased TWS over the southern TP mountains can be attributed primarily to the PME deficit from the southeast North Atlantic with considering optimum lags (Extended Data Figs. 3m-n).

Comment 17

Line 463, which ERA5 reanalysis data?

Reply: Here, the ERA5 reanalysis data denotes the ERA5-based snow cover area over the region r during 2003-2016 (Lines 593-595).

Lines 593-595 in revised description

The models are firstly trained and validated based on the GRACE-based TWS or the ERA5-based snow cover area from over region r during 2003-2016

Comment 18

There is no conclusions?

Reply: Thanks for the suggestion. According to the “Guides for the Author” of Nature, there is no requirement on the Conclusion section but strict word limitation (less than 3200) on the Main Text. Since there are limited rooms for Conclusion section, we collectively presented the main findings of the study in the Abstract section. Thanks again for these highly constructive comments!

3. Appendix: Figures

Figure 1. Diagnosis of water vapor into the TP. (a) The clustered trajectories from source regions to TP during 2003-2017. (b) Particle-scale water vapor RCs from all source regions to the TP. (c) RCs of water vapor from the first to fourth source regions. (d-e) Combined RCs (RC_{main}) for Asia, India Ocean, North Africa and North Atlantic as compared to the total RC of all considered source regions (RC_{full}). The month count denotes the number of months when RC_{main}/RC_{full} is in each interval. The continental map data hereafter is based on country-scale world map data³¹. The Tibet Plateau map data³² hereafter is acquired from the National Tibetan Plateau Data Center, China. The figure is plotted using R³³. And results used to generate the figure are available in Zenodo³⁴.

Supplementary Figure 1. Backward trace analysis for SR1-2 and TP. (a,c,e) Clustered trajectories into the SR1 (a), SR2 (c) and TP (e). (b,d,f) denote the monthly relative contributions of water vapor from main source regions to the SR1 (b), SR2 (d) and TP (f) when the on-route changes are disregarded. The boxplot margins mark the minimum, 25% quantile, median, 75% quantile and maximum value of the monthly relative contributions during 2003-2017. The continental world map data³¹ and map data of Tibet Plateau³² in panels a, c and e are acquired from public data source and plotted by R³³. And results used to generate the figure are available in Zenodo³⁴.

Figure 2. Anomalous TWS increase over central TP. (a-b) Spatial pattern of the modified Mann-Kendall trends of TWS across Eurasia and PME over oceans. (c) Cross-correlations between PME over North Atlantic and TWS over HSRs across Eurasia and the southern TP mountains (TPM1-2). (d) Temporal variations in PME over southeast North Atlantic (NATO3) and average TWS across the HSRs. (e) Temporal variations in PME over NATO3 and TWS in TPMs and central TP. (f) Variation in TWS amplitude during 2003-2016. Hereafter, the TPM1-2 refers to southwest and southeast TP mountains, respectively. The TPS refers to the central TP surface and the NTP refers to northern TP. The continental map data³¹ and the map data of TP³² in panels a-b are from the public data source and plotted using R³³. And results used to generate the figure are available in Zenodo³⁴.

Supplementary Figure 4. Relations between PME over oceans and PME and TWS over SRs. (a-c) denote spatial locations of the subregions for North Atlantic (a), Indian Ocean (b) and mid-latitude Eurasia along water vapor trajectories to TP (c). (d-e) denote relations between PME over oceans and PME over SR1-3. (g-i) denote relations between PME over oceans and TWS over SR1-3. The continental world map data³¹ in panels a-c and the map data of Tibet Plateau³² in panel c are acquired from public data source and plotted by R³³. And results used to generate the figure are available in Zenodo³⁴.

Supplementary Figure 5. Relations between PMEs and TWSs in SR1-3. (a-c) refer to the temporal variations in PMEs and TWSs in SR1-3, respectively. The “cor” in the plot refers to the correlation coefficient hereafter. And results used to generate the figure are available in Zenodo³⁴.

Supplementary Figure 6. Cross correlations between PME over North Atlantic and TWS in lands. (a) refers to the spatial locations of HSR1-12 and LSR1-2. (b) refers to the relations between TWSs in HSR1-12 and LSR1-2 and PMEs in regions of North Atlantic and Indian Ocean. The TPS in the plot refers to the central TP surface. The continental world map data³¹ and map data of Tibet Plateau³² in panel a are acquired from public data source and plotted by R³³. Here, the sub-region where the cross-correlation coefficients >0.4 is defined as the HSR. Rest of sub-region is defined as the LSR. And results used to generate the figure are available in Zenodo³⁴.

Supplementary Figure 10. Air temperature over oceans. (a) refers to the variation in the regional average air temperature over the Indian Ocean. (b) refers to the variation in the regional average air temperature over the south (including southeast and southwest) North Atlantic. And results used to generate the figure are available in Zenodo³⁴.

Supplementary Figure 12. Relative contribution of water vapor based on moisture gain from source regions. (a-b) refer to the relative contributions without (a) and with (b) considering the on-route changes. And results used to generate the figure are available in Zenodo³⁴.

Supplementary Figure 24. Determination of the optimum number of the clustered trajectories.

(a) is the gap statistic k value for the determination of the optimum number of the clustered trajectories. (b-d) refer to the clustered trajectories when the optimum number of the clustered trajectories is 3. The continental world map data³¹ and map data of Tibet Plateau³² in panels **b-d** are acquired from public data source and plotted by R³³. And results used to generate the figure are available in Zenodo³⁴.

Supplementary Figure 25. Clustered trajectories into TP. The continental world map data³¹ and map data of Tibet Plateau³² in all panels are acquired from public data source and plotted by R³³. And results used to generate the figure are available in Zenodo³⁴.

Extended Data Figure 2. Attribution analysis of the TWS in the central TP. (a, c) temporal variances in TWS and PME in central TP and TWSs in southwest (TPM1) and southeast (TPM2) TP mountains. (b, d) include coefficients $a1$, $b1$, $a2$ and $b2$ of the linear regression models in (a, c). And results used to generate the figure are available in Zenodo³⁴.

Extended Data Figure 3. PME over the NATO3 and continental TWS. (a-l) refer to comparisons between PME over the NATO3 and TWS in HSR1-12. (m-n) are same as panels a-l but for TWS in the southern TP mountains (TPM1-2). (o) is the same as panels a-l but for TWS in the TPS. The “ccor” in the plot indicates cross-correlation coefficient and the numbers in the parenthesis present the p values in cross-correlation analysis. And results used to generate the figure are available in Zenodo³⁴.

Extended Data Figure 4. Illustration of the blocking effect by HMA. (a) Mechanism of the interception of the eastward propagation of the PME deficit in North Atlantic by southern HMA. (b) Variation in the interception of the eastward propagation of the PME deficit in North Atlantic by southern HMA. The southwest (TPM1) and southeast (TPM2) TP mountains are all in southern HMA. The terrain texture in panel a is acquired from Google and the open-free heightmap (<https://tangrams.github.io/heightmapper/>) plotted by the authorized Adobe Photoshop 2021 plugin named “3D map generator - terrain”. The panel b is plotted using authorized Adobe Illustrator 2021. And this figure is based on the result of this study³⁴.

Extended Data Figure 6. Standardized annual snow cover area and the area under positive annual TWS in TP. The SC_Moun1 and SC_Moun2 refer to the standardized snow cover area in TPM1 and TPM2, respectively. The PosTWS_Area_TP denotes the standardized area under positive annual TWS in TP during 2003-2016. And results used to generate the figure are available in Zenodo³⁴.

Extended Data Figure 7. Evolution in the area under negative annual TWS in TP. (a-n) Spatiotemporal transition pattern of the border of the area affected by negative annual sum of the TWS in Tibet Plateau from 2003-2016. (o) the variation in the area in TP affected by positive and negative annual sum of the TWS. The map data of Tibet Plateau³² in panels a-n is acquired from public data source and plotted by R³³. And results used to generate the figure are available in Zenodo³⁴.

Response to Reviewer 2

1. Summary comments and reply

Review of “Sustainability of Asian water tower threatened by drying and warming ocean”, by Zhang, Shen et al.

The authors present a very comprehensive study to possible explanations for the spatial variations in trends of terrestrial water storage on the Tibetan Plateau. They use a combination of the FLEXPART particle dispersion model with reanalysis climate data and GRACE remotely sensed TWS data, and argue that TWS decline in the southern TP can be attributed to a deficit in precipitation minus evapotranspiration from the southeast North Atlantic. They further argue that the high mountain ranges in Asia block the propagation of the deficit further onto the central TP which leads to increasing TWS there. Lastly, they demonstrate using CMIP6 GCM data how this blocking could weaken in the future, causing also the central TP to witness future TWS deficits.

To my knowledge this is an original approach, it is mostly well described and components of the analysis thoroughly validated. The findings are of broad interest and provide new knowledge about the possible mechanisms of variability in climate change trends across the TP. In some cases, I am not fully convinced about the robustness of intermediate conclusions, which propagate into the further analysis. I believe these need further clarification. These are included in the specific comments below.

Reply: Thank you for your positive comments on our work. We have thoroughly checked the manuscript and revised it based on your comments and concerns. To make it easy to follow, we have included appropriate references to the revised manuscript (e.g., line numbers). Quoted text from the manuscript is provided in *italics*.

2. Specific comments

Comment 1

Title: The title seems somewhat strange in that it could be read as that the ocean is drying up. Consider rewording.

Reply: Thank you for your suggestion. We have modified the title as “Changes in oceanic climate threaten the sustainability of Asian water tower”. Hopefully, the modified title better reflects the contents and is free of misunderstanding.

Comment 2

L84-85: I do not understand the statement ‘Since glacier melting dominated the TWS changes in the high mountains instead of elevation’.

Reply: Thank you for the comment. It should be the “Since glacier melting dominated the TWS changes in the HMA¹⁹”. We have improved this description in the revised version of the manuscript as follows.

Lines 88-90 in the revised manuscript

Since glacier melting dominated the TWS changes in the HMA¹⁹, we opine that shrinking glaciers and snow cover can modulate the thermodynamic conditions in the HMA and the resultant damping of the blocking.

Comment 3

Figure 1 a+b: The water vapor trajectories are hard to distinguish. I suggest to try more contrasting colors and/or different line types.

Reply: We have distinguished the trajectories in Figure 1 a+b with different colors and different line types. Besides, Supplementary Figure 25 includes the trajectory in each panel to clearly display the trajectories. Determination of the clustered trajectories sees Supplementary Note 1.

Supplementary Note 1

Based on the determination standard⁵⁶ for the optimal number of the clusters using k-means method, we initially determined the optimal number of the clusters for the trajectories is 3, and we further derived the 3 clusters of the trajectories from all trajectories (Supplementary Fig. 24). It suggests that there are only 3 eastward trajectories into TP without denoting the trajectory from the Indian Ocean (Supplementary Fig. 24). Although 3 is the optimal number of clusters, the result could not illustrate enough information about the transition path from the source regions to the TP. As for the criterion for determining the optimum number of clustered trajectories, we do not find relative standards in the handbook of the FLEXPARTv10.4 model and we also find various number of clusters used in other studies^{47,51}. Thus, the optimum number of clusters is generally determined on a case-by-case basis.

Thus, in this study, we attempt to enrich the information of the trajectories and avoid far overlapping of the trajectories at same time. And we set the number of the clusters as 20, and derive 20 clustered trajectories (Supplementary Fig. 25). However, the 9th trajectory has an abnormal turn from East Asia to TP, attributed to the disturbance induced by the zero-longitude centered map projection on the process of clustering trajectories. Since the rest of 19 trajectories are enough for illustrating the transition path from source regions to TP, thus, we finally only keep the 19 clustered trajectories in the main text.

Comment 4

Figure S1: Suggest to change the y-axis for panels b-d to be able to see more details, like also done in other rows in the figure.

Reply: Thank you for the comment. We changed the y-axis scale fitting all panels to display more details in Figure S1 and Figure S2.

Comment 5

L139-140: Looking at Figs S1 and S2, the contributions from North Africa are about equal to the North Atlantic. Therefore, I wonder how the authors exactly come to the conclusion that contributions from the North Atlantic dominate over contributions from North Africa? This conclusion is important as it propagates further in the analysis when focus is going to the North Atlantic region. I would guess that the ocean being a body of water supplies more water vapor than the northern African land mass with the dry Sahara, but I cannot deduct that from Figs S1-S2. In Figs S3-S9 correlations for the North Atlantic and Indian Ocean are shown and North Africa is not further investigated whereas the data in Figs S1 and S2 suggest it should. I am no expert in particle dispersion modeling, so the explanation could be something obvious, but then it requires a bit more explanation for the reader. Also, while Figs S3-S9 indeed show better correlations for North Atlantic compared to Indian Ocean, it seems that Indian Ocean still plays a large role, in particular IO4 (Fig S8, panel c).

Reply: We summarize the above questions as two questions and answer them accordingly.

Question 1: Why did not the study focus on the contribution of water vapor from the North Africa and other land source regions?

Answer: It is because that the ocean is the water vapor source into the continent by a large-scale atmospheric circulation. We performed additional backward trace analyses for the particles initially residing over the SR1 and SR2. Since SR3 covers most of area of TP, we did not perform further backward trace analysis for the SR3. According to Supplementary Fig. 1, it demonstrates that *the oceans (e.g., Indian Ocean and North Atlantic) recharge the water vapor of air particles over the lands (e.g., Asia and Africa) that further transit water vapor into the TP via the two main trajectories. The first is from the North Atlantic to the western TP transported by the westerlies (Figure 1a-b), and the other is the intermedia current (see Figure 1a-b) that brings water vapor from*

North Atlantic and Indian Ocean into the southern TP. (see Lines 100-106)

Question 2: Indian Ocean still plays a large role, in particular IO4 according to Figs S3-S9.

Answer: The reason why the PME over Indian Ocean was not included in the further analysis is that the relations between TWS over SR1-3 and PME over IO4 is becoming weaker with the distance closer to TP (from SR1 to SR3). This does not apply to the directions of the two main transition paths (westerlies and intermedia current) into TP (Figure 1b). Thus, we excluded the PME over Indian Ocean in the further analysis. However, we found that the air temperature over the southeast TP mountains (TPM2) are coherent with the air temperature over the Indian Ocean (Extended Data Figure. 5). Thus, in the further analysis, air temperature variations over the Indian Ocean have been included when we developed the models for the projection of the air temperature over the TPM2 (Equations. 18-19).

Comment 6

L198-201: I find the preceding argumentation and interpretation of Extended Data Figures 3 and 4 for the point that the high mountains block the propagation of the PME deficit hard to follow. Therefore, it also does not fully convince me and I am not sure how valid the point is. I suggest to revisit lines 186-201 and refer to particular panels in the referred figures and elaborate explanation for this important part of the paper.

Reply: Thanks for the suggestion. Following your instruction, we have included the particular panels (e.g., Extended Data Figs. 3m-n) to the referred figures and revise the context accordingly in Lines 186-201 (now in Lines 159-181). And we agree with you that the interpretation of origin Extended Data Figs. 3 and 4 (Extended Data Figs. 2 and 3 now) cannot fully support the point that the high mountains block the propagation of the PME deficit. The interpretation of these two Figures in Lines 185-207 is to raise the scientific question as to whether the abnormal increase in TWS over the central TP

could be attributed to the interception of the PME deficit by the high mountains. And the subsequent section “HMA damps the impact of oceanic drying” is applied to demonstrate this point based on the observation and model simulations.

The reason that you feel puzzled here might be the lack of the link between context. To address your concerns and avoid potential misunderstanding by readers, we have included the link at the end of the second section in the revised manuscript (Lines 176-181).

Lines 159-176

Our linear attribution analysis suggests that the increased TWS over the central TP can be attributed to decreased TWS across the southern TP mountains instead of increased PME (Extended Data Fig. 2). And the cross-correlation analysis suggests the decreased TWS over the southern TP mountains can be attributed primarily to the PME deficit from the southeast North Atlantic with considering optimum lags (Extended Data Figs. 3m-n). Besides, significant relations between TWS depletion (e.g., glacier mass loss) and the reductions in snow cover area over southern TP mountains are detected (Supplementary Fig. 7). The latter is mainly caused by regional warming (Supplementary Fig. 8). Even though glacier/snow meltwater from the southern TP mountains could increase TWS in the central TP³⁵, such meltwater also replenishes TWS in the basins surrounding the TP. A comparative analysis for the surrounding basins (HSR8-12) of the TP and the central TP shows that the surrounding regions are directly exposed to the westerlies and intermedia current that largely terminate at the southern margin of the central TP (Figure 2b). Accordingly, the TWS changes in surrounding basins (Extended Data Figs. 3i-l) are consistent with the PME deficit transited by the westerlies from southeast North Atlantic but it is not the case for the TWS in the

central TP (Extended Data Figures 3o).

Lines 176-181

Since the southern TP mountains belong to the HMA, it is reasonable to assume that blocking by the HMA's high topography dampens the propagation of the PME deficit that emerged in the southeast North Atlantic towards the TP, which in turn could have resulted in the observed increase in TWS over the central TP. We will further investigate this assumption in the next section based on the FLEXPART simulation.

Comment 7

L324-327: Why were these particular CMIP6 models included and others not? I also believe MICRO6 is a typo and should be MIROC6.

Reply: The selected CMIP6 models here include ACCESS-ESM1.5, BCC-CSM2-MR, CanESM5, GFDL-ESM4, IPSL-CM6A-LR, MIROC6, MRI-ESM2.0 and NorESM2-LM. The main reason why we select these particular CMIP6 models is that they are developed by the major organizations around the world and have been applied in the previous studies (Gillett et al., 2021; McKinnon et al., 2021). Besides, since the multiple average and weighting sum of CMIP6 outputs (e.g., PME and T) are finally determined for the future projections and organizations always provide several CMIP6 models by themselves, to avoid impacts on the weights in calculations of the weighting sum of CMIP6 outputs, we select only one CMIP6 model from each organization. Additionally, we selected the CMIP6 models under the r1i1p1f1 format. However, not all of CMIP6 models have the format, e.g., GISS-E2-1-G, HadGEM3-GC31-LL and CESM2. The r1i1p1f1 format is also one of the criteria for the selection of the CMIP6 models.

And the MICRO6 is a typo, we have corrected this typo through the whole manuscript.

Reference

Gillett, N.P., Kirchmeier-Young, M., Ribes, A. et al. Constraining human contributions to observed warming since the pre-industrial period. *Nat. Clim. Chang.* 11, 207–212 (2021). <https://doi.org/10.1038/s41558-020-00965-9>

McKinnon, K.A., Poppick, A. & Simpson, I.R. Hot extremes have become drier in the United States Southwest. *Nat. Clim. Chang.* 11, 598–604 (2021). <https://doi.org/10.1038/s41558-021-01076-9>

Comment 8

L491-492 and Figs S18-19: Comparing timeseries of CMIP6 output directly year-to-year to reanalysis data in my experience never provides good correlation. I believe it is more valid to compare climate characteristics averaged over longer periods, like distributions, degree of variability.

Reply: Thanks for the suggestion. Here, we compared the monthly CMIP6 precipitation and temperature with those from the ERA5 reanalysis data. Even though the timeseries of the monthly CMIP6 output alone does not have a good correlation with the ERA5 reanalysis data, multi-weight CMIP6 output based on outputs from multiple CMIP6 outputs could improve the correlations with ERA5 reanalysis data. Thus, to improve the robustness of the projected results, we applied multiple mean and weighting sum of the CMIP6 outputs for the futural projection.

Besides, by following your suggestion, we compared the distributions of the CMIP6 outputs and the distribution for the ERA5 reanalysis data using the Kolmogorov-Smirnov test (Supplementary Fig. 16). It demonstrates that distributions of outputs from CMIP6 do not have significant bias from the distribution of the ERA5 reanalysis data when $p < 0.05$. Given that the projected model is based on the CMIP6 output series, we determined both multiple mean and weighting sum of CMIP6 outputs to improve the capability of the CMIP6 models on reflecting the temporal variations in precipitation and temperature.

Comment 9

Supplementary Table 1: I suggest to omit this table, the statement in the Methods section is sufficient.

Reply: Thanks for the suggestion. We have removed the Table in the revised Supplementary Information file.

Comment 10

A recent relevant paper that was published in the meantime to consider citing where appropriate:

Li, X., Long, D., Scanlon, B. R., Mann, M. E., Li, X., Tian, F., Sun, Z., & Wang, G. (2022). Climate change threatens terrestrial water storage over the Tibetan Plateau. *Nature Climate Change*. <https://doi.org/10.1038/s41558-022-01443-0>

Reply: Thanks for the note. We cited this relevant paper in the main text as the 20th reference. Different from this relevant paper, we explored and clarified the mechanism behind the heterogeneities in the trends in terrestrial water storage (TWS) over the TP from the perspective of the large-scale atmospheric circulation. In our study, we evidenced that the blocking effect of the westerlies-carried changes in ocean climates by the High Mountain Asia as the cause for the abnormal increase in TWS. In addition, the blocking effect is demonstrated in weakening. Further, we also projected the area under TWS deficit in TP in future by considering the weakening blocking effect by the High Mountain Asia and oceanic climate changes.

Lines 73-75

The unprecedented changes in the HMA's water systems have been reported in numerous studies^{3,8,19,20}. However, the atmospheric mechanisms causing the distinct and regionally-varying TWS changes are not well understood⁵.

At the end, thanks for your time in reviewing our work and providing insightful suggestions.

3. Appendix: Figures

Figure 1. Diagnosis of water vapor into the TP. (a) The clustered trajectories from source regions to TP during 2003-2017. (b) Particle-scale water vapor RCs from all source regions to the TP. (c) RCs of water vapor from the first to fourth source regions. (d-e) Combined RCs (RC_{main}) for Asia, India Ocean, North Africa and North Atlantic as compared to the total RC of all considered source regions (RC_{full}). The month count denotes the number of months when RC_{main}/RC_{full} is in each interval. The continental map data hereafter is based on country-scale world map data³¹. The Tibet Plateau map data³² hereafter is acquired from the National Tibetan Plateau Data Center, China. The figure is plotted using R³³. And results used to generate the figure are available in Zenodo³⁴.

Figure 2. Anomalous TWS increase over central TP. (a-b) Spatial pattern of the modified Mann-Kendall trends of TWS across Eurasia and PME over oceans. (c) Cross-correlations between PME over North Atlantic and TWS over HSRs across Eurasia and the southern TP mountains (TPM1-2). (d) Temporal variations in PME over southeast North Atlantic (NATO3) and average TWS across the HSRs. (e) Temporal variations in PME over NATO3 and TWS in TPMs and central TP. (f) Variation in TWS amplitude during 2003-2016. Hereafter, the TPM1-2 refers to southwest and southeast TP mountains, respectively. The TPS refers to the central TP surface and the NTP refers to northern TP. The continental map data³¹ and the map data of TP³² in panels a-b are from the public data source and plotted using R³³. And results used to generate the figure are available in Zenodo³⁴.

Supplementary Figure 1. Backward trace analysis for SR1-2 and TP. (a,c,e) Clustered trajectories into the SR1 (a), SR2 (c) and TP (e). (b,d,f) denote the monthly relative contributions of water vapor from main source regions to the SR1 (b), SR2 (d) and TP (f) when the on-route changes are disregarded. The boxplot margins mark the minimum, 25% quantile, median, 75% quantile and maximum value of the monthly relative contributions during 2003-2017. The continental world map data³¹ and map data of Tibet Plateau³² in panels a, c and e are acquired from public data source and plotted by R³³. And results used to generate the figure are available in Zenodo³⁴.

Supplementary Figure 2. Monthly contribution rates accounting for on-route changes. Panels “(a)” to “(m)” refer to the contribution rates of water vapor during 2003-2017 from Asia (AS), Indian Ocean (IO), North Africa (NAF), North Atlantic (NATO), Mediterranean Sea (MS), Europe (EU), Pacific Ocean (PO), Red Sea (RS), Tibet Plateau, North America (NAM), Caspian Sea (CS), Black Sea (BS), and Arctic Ocean (AO). And results used to generate the figure are available in Zenodo³⁴.

Supplementary Figure 3. Monthly contribution rates disregarding on-route changes. Panels “(a)” to “(m)” refer to the contribution rates during 2003-2017 from AS, IO, NAF, NATO, MS, EU, PO, RS, TP, NAM, CS, BS, and AO, respectively. And results used to generate the figure are available in Zenodo³⁴.

Supplementary Figure 4. Relations between PME over oceans and PME and TWS over SRs. (a-c) denote spatial locations of the subregions for North Atlantic (a), Indian Ocean (b) and mid-latitude Eurasia along water vapor trajectories to TP (c). (d-e) denote relations between PME over oceans and PME over SR1-3. (g-i) denote relations between PME over oceans and TWS over SR1-3. The continental world map data³¹ in panels a-c and the map data of Tibet Plateau³² in panel c are acquired from public data source and plotted by R³³. And results used to generate the figure are available in Zenodo³⁴.

Supplementary Figure 7. TWS and snow cover in the southern TP mountains. (a-b) refer to the temporal variation in TWSs and snow cover in the southwest (TPM1) and southeast (TPM2) TP mountains. And results used to generate the figure are available in Zenodo³⁴.

Supplementary Figure 8. Relations between snow cover and negative temperature in TPMs. (a-b) refer to the comparison between the snow cover and negative values of the temperature in the southwest (TPM1) TP mountain. (c-d) refer to the comparison between in southeast (TPM2) TP mountain. The negative temperature equals to -1 multiplies temperature. And results used to generate the figure are available in Zenodo³⁴.

Supplementary Figure 16. Comparison of the cumulative distributions between CMIP6 output and ERA5. (a-c) denote the comparison of the cumulative distribution between CMIP6 and ERA5 PME over southeast North Atlantic during 2003-2016 under historical-SSP245 (a) and historical-SSP585 (b) scenarios. (d-f) denote the comparison between CMIP6 and ERA5 temperature over southwest TP mountain during 2003-2016 under historical-SSP245 (d) and historical-SSP585 (e) scenarios. (g-i) are same as (d-f) but for temperature over southeast TP mountains. Here, the Kolmogorov-Smirnov test is applied to evaluate the distance between two contributions. And results used to generate the figure are available in Zenodo³⁴.

Supplementary Figure 24. Determination of the optimum number of the clustered trajectories. (a) is the gap statistic k value for the determination of the optimum number of the clustered trajectories. (b-d) refer to the clustered trajectories when the optimum number of the clustered trajectories is 3. The continental world map data³¹ and map data of Tibet Plateau³² in panels **b-d** are acquired from public data source and plotted by R³³. And results used to generate the figure are available in Zenodo³⁴.

Supplementary Figure 25. Clustered trajectories into TP. The continental world map data³¹ and map data of Tibet Plateau³² in all panels are acquired from public data source and plotted by R³³. And results used to generate the figure are available in Zenodo³⁴.

Extended Data Figure 2. Attribution analysis of the TWS in the central TP. (a, c) temporal variances in TWS and PME in central TP and TWSs in southwest (TPM1) and southeast (TPM2) TP mountains. (b, d) include coefficients $a1$, $b1$, $a2$ and $b2$ of the linear regression models in (a, c). And results used to generate the figure are available in Zenodo³⁴.

Extended Data Figure 3. PME over the NATO3 and continental TWS. (a-l) refer to comparisons between PME over the NATO3 and TWS in HSR1-12. (m-n) are same as panels a-l but for TWS in the southern TP mountains (TPM1-2). (o) is the same as panels a-l but for TWS in the TPS. The “ccor” in the plot indicates cross-correlation coefficient and the numbers in the parenthesis present the p values in cross-correlation analysis. And results used to generate the figure are available in Zenodo³⁴.

Extended Data Figure 5. Maximum Covariance Analysis between oceanic and continental T. (a) the spatial coefficients of the air temperature (T) over ocean and that across Eurasia under the first leading mode. (b) the time coefficients of the air temperature over ocean and that across the Eurasia under the first leading mode with the explained variance ratio as 27.66%. (c) temporal variation in air temperature over southwest North Atlantic (NATO1) and air temperature in southwest TP mountains (TPM1). (d) temporal variation in air temperature in northwest Indian Ocean (IO2) and air temperature in southeast TP mountains (TPM2). The continental world map data³¹ and map data of Tibet Plateau³² in the panel a are acquired from public data source and plotted by R³³. And results used to generate the figure are available in Zenodo³⁴.

Reviewer Reports on the First Revision:

Referees' comments:

Referee #1 (Remarks to the Author):

The authors have given a correct explanation to all my questions, in addition to carrying out new calculations to improve the study and meet the objectives. So, in my opinion, the new version is correct for publication, thank you.

Referee #2 (Remarks to the Author):

The authors made good use of the feedback provided by the reviewers and the manuscript has improved. Most of my concerns have been addressed, although to me still much is unclear or unconvincing in the section 'Blocking effects of the HMA'. More about this is in the specific comments below.

Note: Line numbers refer to the Track Changes version of the revised manuscript.

L93: Suggestion to change 'Diagnosis' to 'Analysis'

L96: Spell out RC at first instance.

L162-164: The correlation observed is clear, but I am not certain to which extent there is really a case of attribution. Could there be another mechanism that causes the correlation?

L179-182: How reasonable this assumption of blocking is remains rather vague to me. Is it not more likely that the TWS loss is mainly driven by increasing temperatures and the TWS in the central TP behaves differently because there are mostly endorheic basins and they do not drain the water out, in contrast to the surrounding areas, where loss of water stored in glaciers/snow drains out to the sea? I acknowledge there is not a straightforward answer and the exploration of the mechanism explored in the paper is very useful, I am just not completely convinced by the paper that this is indeed the driving mechanism. I suggest to use very careful wording that this could be one possible mechanism.

L209-211: It remains unclear what is specifically meant by 'alters the local thermodynamic condition accordingly', and how it further leads to the variation in the blocking effect. As this seems crucial to substantiate the hypothesis of the blocking effect, this needs to be convincingly specified.

L216: The snow cover area cannot decline by more than 100%. This is probably related to the use of values of indices and confusing. Similar issue in L220-221. It is explained in the manuscript and Methods that these large percentual changes are because of use of indices, but still it is confusing. I suggest to think of an alternative way than percentual changes in index values, to indicate the changes.

L217-219: It is unclear to me what is demonstrated here. Snow cover decreases in TPM1 and TPM2. But what is specifically the link with weakening of the blocking effect?

Extended Data Figure 6: It is confusing that now the suffixes _Moun1 and _Moun2 are used where TPM1 and TPM2 were used before (and also in the caption).

L223: It remains unclear what is meant by the corridor. Is it equal to all the areas with negative TWS? Or a specific atmospheric trajectory? Please point to the specific panel(s) where this is seen, or

indicate it as corridor in one of the figures.

L246-247: Here also I think specification is required for the reasoning why accelerated melting weakens the blocking effect.

Author Rebuttals to First Revision:

Response to Reviewer 2

Referee #2 (Remarks to the Author):

The authors made good use of the feedback provided by the reviewers and the manuscript has improved. Most of my concerns have been addressed, although to me still much is unclear or unconvincing in the section ‘Blocking effects of the HMA’. More about this is in the specific comments below.

Note: Line numbers refer to the Track Changes version of the revised manuscript.

Reply: Thank you for your positive comments, which are very helpful to further improve the manuscript. Your remaining specific comments are addressed below. Note that all Figures mentioned in the Reply can be found at the end of this document.

Comment 1

L93: Suggestion to change ‘Diagnosis’ to ‘Analysis’

Reply: We replaced “Diagnosis” with “Analysis” in Line 90. The revised title of the first section is now “*Analysis of water vapor into the TP*”.

Comment 2

L96: Spell out RC at first instance.

Reply: We have spelled out RC as to be “relative contribution” in this revision:

Lines 93-94 in the Track Changes version of the revised manuscript.

... we find long-term average relative contributions (RCs) of ~42% from Asia ...

Comment 3

L162-164: The correlation observed is clear, but I am not certain to which extent there is really a case of attribution. Could there be another mechanism that causes the correlation?

Reply: The comment was related to the sentence (L162-164) “*Our linear attribution analysis suggests that the increased TWS over the central TP can be attributed to decreased TWS across the southern TP mountains instead of increased PME (Extended Data Fig. 2).*”. We agree that the observed correlation could also be due to other mechanisms (see our answer to the Question 2, Comment 4) and that the word “attribution” might have been too strong in this case. In the revised version, we thus choose a more careful wording (Lines 154-157):

Lines 154-157 in the Track Changes version of the revised manuscript

Our linear attribution analysis suggests that the increased TWS over the central TP might be related to decreased TWS across the southern TP mountains instead of local increased PME (Extended Data Fig. 2).

Comment 4

L179-182: How reasonable this assumption of blocking is remains rather vague to me. Is it not more likely that the TWS loss is mainly driven by increasing temperatures and the TWS in the central TP behaves differently because there are mostly endorheic basins and they do not drain the water out, in contrast to the surrounding areas, where loss of water stored in glaciers/snow drains out to the sea? I acknowledge there is not a straightforward answer and the exploration of the mechanism explored in the paper is very useful, I am just not completely convinced by the paper that this is indeed the driving mechanism. I suggest to use very careful wording that this could be one possible mechanism.

Reply: Thank you for these critical comments. We answer them on the basis of the two main questions here below.

Question 1: Is it not more likely that the TWS loss is mainly driven by increasing temperatures?

Reply: We concur that increasing temperatures in the southern TP mountains have an impact on the regional TWS, and indeed, we do consider “temperature” as one of the variables for predicting the future TWS in the region (see Equation 17). However, our analysis suggests that this is not the main driving factor. Although the warming induced a decrease in the seasonal snow cover (Supplementary Fig. 8a,e), we noticed that TWS has only a poor relation to the temperature in the southern TP mountains (Supplementary Figure 8c,g). This is because by definition, TWS is the sum of the surface water, groundwater, soil moisture, canopy water and water stored in glacier ice. The reason why temperature is not the main driving factor is that glacier melt alone is not representative for the total TWS variations, thus explaining the poor correlations between TWS and temperatures in the southern TP mountains (Supplementary Figure 8c,g). This consideration also explains why we could not draw the conclusion that increasing temperatures mainly drive the decrease in TWS.

In our revision (Line 160-163), we better point out that the observed snow cover decrease is indeed induced by warming but that changes in TWS are less clearly affected by that (Supplementary Figure 8):

Lines 160-163 in the Track Changes version of the revised manuscript

We also detect a significant relation between TWS depletion (e.g., glacier mass loss) and the reductions in snow cover area over the southern TP mountains (Supplementary Fig. 7). In contrast to TWS declines, the reduction in snow cover area is mainly caused by regional warming (Supplementary Fig. 8).

Question 2: Is it not more likely that the TWS in the central TP behaves differently because there are mostly endorheic basins and they do not drain the water out, in contrast to the surrounding areas, where loss of water stored in glaciers/snow drains out to the sea?

Reply: We concur with this possibility. However, some of our results are in contradiction to this. Unlike the increased TWS observed over endorheic basins in the central TP (Figure 2b and Extended Data Figure 3o), the TWS over the Tarim basin in HSR12 (the largest endorheic basin in China) declines despite of the same meltwater supply from the southern TP mountains as for the central TP (Figure 2b and Extended Data Figure 3l). This contrasting behavior suggests that the endorheic nature of the basins is not sufficient for explaining the observed increase in TWS. That said, we now provide a more balanced discussion about this, and instead of presenting a change in blocking effects as the sole hypothesis, we also discuss the potential for endorheic basins to play a role in the TWS variation. This additional discussion is found at Lines 165-184 of the revised manuscript:

Lines 165-184 in the Track Changes version of the revised manuscript

The endorheic character of the central TP differs from the exorheic nature of the basins situated in the south of the TP (e.g., Indus basin in HSR11, Figure 2b). This difference could potentially allow for the central TP to accumulate meltwater from the southern TP mountains, thus explaining the observed increase in TWS. However, the fact that the TWS over the largest endorheic basin in China (the Tarim basin in HSR12; Figure 2b) declines despite of the same meltwater supply from the southern TP mountains as for the central TP, seems to rule out this hypothesis. A comparative analysis for the basins surrounding the central TP (HSR8-12), moreover, shows that the surrounding regions are directly exposed to the westerlies and intermediate currents that largely terminate at the southern margin of the central TP (Figure 2b). This means that the TWS changes in the surrounding basins (Extended Data Figs. 3i-l) are consistent with a propagation of the PME deficit by the westerlies from southeast North Atlantic, while this is not the case for the TWS in the central TP (Extended Data Fig. 3o). Since the southern TP mountains belong to HMA, we suggest that the abnormal increase in TWS in the central TP could be linked to changes in blocking effects caused by HMA's high topography. In particular, this topography dampens the propagation of the PME

deficit that emerge in the southeast North Atlantic, and we hypothesize that a reduction in the propagation of the PME deficit could have resulted in the observed increase in TWS over the central TP.

Comment 5

L209-211: It remains unclear what is specifically meant by ‘alters the local thermodynamic condition accordingly’, and how it further leads to the variation in the blocking effect. As this seems crucial to substantiate the hypothesis of the blocking effect, this needs to be convincingly specified.

Reply: For wind systems across mountains, two types of air flow are often distinguished^{36,37}: terrain-driven flow and thermal-driven flow. The latter depends on the conditions of the internal, thermal boundary layers which are formed by differences in land surface temperatures. Such differences can result, for example, from differences in the snow cover fractions over high mountains. More specifically, the solar radiation on snow-covered surfaces leads to a stable thermal internal boundary layer which in turn produces negative air buoyancy and thus katabatic wind. In contrast, the solar radiation over snow-free area leads to a convective thermal internal boundary layer. This causes positive buoyancy thus initiating upslope wind. For the air flows into the TP, more intense katabatic winds induced by a more extensive snow cover, increases the difficulty for air flows to cross the HMA, thus enhancing the mountains’ blocking effect. In contrast, the formation of upslope winds caused by a reduction in snow cover area decreases could assist air flow across HMA’s main mountain ridges, thus weakening their blocking effect.

In the revision, we provided additional information on what we meant:

Lines 205-218 in the Track Changes version of the revised manuscript

Regional warming in the southern TP has led to a reduction in glacier and snow cover areas. Studies dedicated to atmospheric flows in mountainous terrain^{36,37} indicate that the solar radiation impacting snow-free areas leads to convective thermal internal boundary layers, which in turn give rise to positive buoyancy and thus upslope wind. In contrast, negative buoyancy is promoted over

snow- and ice-covered areas, inducing katabatic, downslope winds. Here, we suggest that the increased occurrence of upslope winds resulting from a reduction of the snow cover area could have assisted air flow across the HMA main mountain ridges, thus weakening their blocking effect. This weakening of the blocking effect could in turn have resulted in an increased propagation of PME deficits into the northern TP, thus contributing to explain the observed TWS decline in the northern TP (Supplementary Figs. 9). Such a mechanism would also help explaining the correlation identified between (i) TWS changes in the northern TP and (ii) the variations in temperature and snow cover area in the southwest TP mountains (Supplementary Figs. 9b-c).

Reference in this reply:

36. Mott, R., Daniels, M., and Lehning, M. (2015). Atmospheric flow development and associated changes in turbulent sensible heat flux over a patchy mountain snow cover. *J. Hydrometeor.* 16, 1315–1340. doi: 10.1175/JHM-D-14-0036.1
37. Sauter, T., and Galos, S. P. (2016). Effects of local advection on the spatial sensible heat flux variation on a mountain glacier. *Cryosphere* 10, 2887–2905. doi: 10.5194/tc-10-2887-2016

Comment 6

L216: The snow cover area cannot decline by more than 100%. This is probably related to the use of values of indices and confusing. Similar issue in L220-221. It is explained in the manuscript and Methods that these large percentual changes are because of use of indices, but still it is confusing. I suggest to think of an alternative way than percentual changes in index values, to indicate the changes.

Reply: We agree that values larger than 100% are confusing in this context. The reason for such values was that we originally determined the relative variation for a standardized index by dividing the variation in the index with the value at the beginning (Eq. 1 in the reply). In our revision, we modified the calculations by determining the relative variation of a standardized index based on the maximum amplitude of all standardized indices (Eq. 2 in the reply). In this way, the relative variations are constrained in the range 0~100%. The new method of calculation, which affects Lines 149-153, 220-222, 223-225, and 229-230, is explained at Lines 547-554 in the Tracked Changes version of the revised manuscript, while the relevant equations read:

$$ORV_{m,n} = \frac{a_m - a_n}{a_n} \times 100\% \quad (\text{Eq. 1})$$

$$MRV_{m,n} = \frac{a_m - a_n}{maxV} \times 100\% \quad (\text{Eq. 2})$$

where a_m and a_n denote the values at moments m and n in a time series x . $ORV_{m,n}$ and $MRV_{m,n}$ are the original and modified relative variations between a_m and a_n . The $maxV$ denotes the maximum amplitude of all standardized indices in this study. Since the maximum scale for all standardized indices is within the range $-3 \sim 3$, thus, we applied the $maxV$ as 6 here.

Lines 149-153 in the Track Changes version of the revised manuscript

We detect a decrease in monthly TWS by $\sim 1\text{Gt}$ ($\sim 47\%$) over the southern TP mountains during 2003-2016, which we attribute to the propagation of a PME deficit from the southeast North Atlantic (Fig. 2e-f), but a synchronous increase in monthly TWS by $\sim 0.5\text{Gt}$ ($\sim 41\%$) in the central TP (Fig. 2e-f).

Lines 220-222 in the Track Changes version of the revised manuscript

Further, we detect an annual TWS decrease by $\sim 11\text{Gt}$ ($\sim 66\%$, or $\sim 13 \text{ kg/m}^3$) and $\sim 12\text{Gt}$ ($\sim 55\%$, or $\sim 12\text{kg/m}^3$) in the southwest and southeast TP mountains during 2005-2016, respectively (Fig. 3b).

Lines 223-225 in the Track Changes version of the revised manuscript

Indeed, the annual average snow cover areas decline by $\sim 11 \times 10^4 \text{ km}^2$ ($\sim 70\%$) and $\sim 7 \times 10^4 \text{ km}^2$ ($\sim 64\%$) in these two regions, respectively (Extended Data Fig. 6).

Lines 229-230 in the Track Changes version of the revised manuscript

Indeed, the area of the TP affected by a TWS deficit (i.e., the area with $TWS \leq 0$) increases by $\sim 167 \times 10^4 \text{ km}^2$ ($\sim 62\%$) (Fig. 3b) during 2005-2016.

Lines 547-554 in the Track Changes version of the revised manuscript

The relative variation for a standardized index is calculated by dividing the index variation with the maximum amplitude of all standardized indices (see Equation 9)

Comment 7

L217-219: It is unclear to me what is demonstrated here. Snow cover decreases in TPM1 and TPM2. But what is specifically the link with weakening of the blocking effect?

Reply: For the connection between snow cover decrease and blocking effects, please see our previous answer to Comment 5. For the specific sentence (now Lines 223-227) we adjusted the wording to:

Lines 223-227 in the Track Changes version of the revised manuscript

Indeed, the annual average snow cover areas decline by $\sim 11 \times 10^4 \text{ km}^2$ ($\sim 70\%$) and $\sim 7 \times 10^4 \text{ km}^2$ ($\sim 64\%$) in these two regions, respectively (Extended Data Fig. 6). We suggest that the reductions in snow cover area and TWS initiate the upslope wind in HMA and thus weakens the blocking effect by assisting air flows (i.e., westerlies) to cross the HMA.

Comment 8

Extended Data Figure 6: It is confusing that now the suffixes _Moun1 and _Moun2 are used where TPM1 and TPM2 were used before (and also in the caption).

Reply: Thank you for your professional suggestion. We now replaced “_Moun1” and “_Moun2” with “TPM1” and “TPM2”, respectively, in the Extended Data Figure 6 and its legend.

Comment 9

L223: It remains unclear what is meant by the corridor. Is it equal to all the areas with negative TWS? Or a specific atmospheric trajectory? Please point to the specific panel(s) where this is seen, or indicate it as corridor in one of the figures.

Reply: Yes, the corridor is defined as the path along which the area with negative TWS propagates from the southern TP mountains to the central TP. In the revision, we have denoted the corridor in the Figure 3e-h with arrows, and defined it in the legend of Figure 3. We also rephrased the description of the corridor in the revised manuscript (Lines 232-234):

Lines 232-234 in the Track Changes version of the revised manuscript

We also find corridors that act as paths for the area with negative TWS to transit from the southern TP mountains to the central TP (Figure 3e-h). These corridors roughly align with the westerlies or the Indian monsoon (Figure 3e-h).

Comment 10

L246-247: Here also I think specification is required for the reasoning why accelerated melting weakens the blocking effect.

Reply: For the connection between snow cover decrease and blocking effects, please see our previous answer to Comment 5 and Lines 205-218 in the revised manuscript. For the specific location in the text, we added a short discussion (Lines 250-254):

Lines 250-254 in the Track Changes version of the revised manuscript

Influenced by a projected warming of air temperatures over oceans by 1.8~3.9 °C (Supplementary Fig. 10) and by projected climatic drying over the southeast North Atlantic, continued melting of glaciers and snow might contribute to further enhance upslope winds and thus to further decrease the mountains' blocking effects during the period 2020-2098.

Appendix: Figures mentioned in the reply

Figure 1. Analysis of water vapor into the TP. (a) The clustered trajectories from source regions to TP during 2003-2017. (b) Particle-scale water vapor RCs from all source regions to the TP. (c) RCs of water vapor from the first to the fourth source region. (d-e) Combined RCs (RC_{main}) for Asia, India Ocean, North Africa and North Atlantic as compared to the total RC of all considered source regions (RC_{full}). The month count denotes the number of months when RC_{main}/RC_{full} is in each interval. The continental map data hereafter is based on country-scale world map data³¹. The Tibet Plateau map data³² hereafter is acquired from the National Tibetan Plateau Data Center, China. The figure is plotted using R³³. And results used to generate the figure are available in Zenodo³⁴.

Figure 2. Anomalous TWS increase over central TP. (a-b) Spatial pattern of the modified Mann-Kendall trends in TWS across Eurasia and PME over oceans. (c) Cross-correlations between PME over North Atlantic and TWS over HSRs across Eurasia and the southern TP mountains (TPM1-2). (d) Temporal variations in PME over southeast North Atlantic (NATO3) and average TWS across the HSRs. (e) Temporal variations in PME over NATO3 and TWS in TPMs and central TP. (f) Variation in TWS amplitude during 2003-2016. Hereafter, TPM1 and TPM2 refer to southwest and southeast TP mountains, respectively, whereas TPS refers to the central TP surface and NTP refers to northern TP. The continental map data³¹ and the map data of TP³² in panels a-b are from the public data source and plotted using R³³. And results used to generate the figure are available in Zenodo³⁴.

Figure 3. Northward expansion of TWS deficit in TP. (a) Elevation profile across the mid-latitude north hemisphere and eastward and northward water vapor trajectories. (b) Annual sum of the standardized trend items of TWS in the southwest and southeast TP mountains and TWS non-deficit area (PosTWS_Area) in the TP during 2003-2016. (c-d) Spatial patterns of annual sums of the monthly mean PME in the North Atlantic and monthly mean TWS across Eurasia during 2003-2016. (e-h) Spatial patterns of annual sum of TWS during 2003-2004, 2005, 2006-2008 and 2009-2016, respectively. The continental map data³¹ and the map data of TP³² in panels c-h are from public data sources, and plotted using R³³. The background terrain map is from Google map acquired by ggmap³⁸ in R. The results used to generate the figure are available in Zenodo³⁴.

Extended Data Figure 2. Attribution analysis of the TWS in the central TP. (a, c) temporal variances in TWS and PME in central TP and TWSs in southwest (TPM1) and southeast (TPM2) TP mountains. (b, d) include coefficients $a1$, $b1$, $a2$ and $b2$ of the linear regression models in (a, c). The results used to generate the figure are available in Zenodo³⁴.

Extended Data Figure 3. PME over the NATO3 and continental TWS. (a-l) show comparisons between PME over the NATO3 and TWS in HSR1-12. (m-n) are same as panels a-l but for TWS in the southern TP mountains (TPM1-2). (o) is the same as panels a-l but for TWS in the TPS. The “ccor” in the plot indicates cross-correlation coefficient and the numbers in the parenthesis present p values in cross-correlation analysis. The results used to generate the figure are available in Zenodo³⁴.

Extended Data Figure 6. Standardized annual snow cover area and the area under positive annual TWS in TP. The PosTWS_Area denotes the standardized area under positive annual TWS in TP during 2003-2016. The results used to generate the figure are available in Zenodo³⁴.

Supplementary Figure 8. Relations between negative temperature and snow cover and TWS in the southern TP mountains. (a-b, e-f) refer to the comparison between the snow cover and negative values of the temperature in the southwest (a-b) and southeast (e-f) TP mountain. (c-d, g-h) refer to the comparison between the TWS and negative values of the temperature in the southwest (c-d) and southeast (g-h) TP mountains. The negative temperature equals to -1 multiplies temperature. And results used to generate the figure are available in Zenodo³⁴.

Supplementary Figure 9. Comparison between TWS in the north TP and other indices in TPM1. (a-c) refers to the temporal variations in TWS (a), snow cover area (b), T (c) in the southwest TP mountains (TPM1) and TWS in the north TP (NTP). (d) refers to the temporal variation in PME in southeast North Atlantic (NATO3) and TWS in the north TP (NTP). The “cor” denotes relations between two indices in each panel. And results used to generate the figure are available in Zenodo³⁴.